# Inhibition of striatonigral autophagy as a link between cannabinoid intoxication and impairment of motor coordination

Cristina Blázquez[1,2], Andrea Ruiz-Calvo[1,2], Raquel Bajo-Grañeras[1,2], Jérôme M Baufreton[3], Eva Resel[1,2], Marjorie Varilh[4], Antonio C Pagano Zottola[4], Yamuna Mariani[4], Astrid Cannich[4], José A Rodríguez-Navarro[2], Giovanni Marsicano[4], Ismael Galve-Roperh[1,2], Luigi Bellocchio[4]*, Manuel Guzmán[1,2]*

[1]Centro de Investigación Biomédica en Red sobre Enfermedades Neurodegenerativas (CIBERNED), Instituto Universitario de Investigación Neuroquímica (IUIN) and Department of Biochemistry and Molecular Biology, Complutense University, Madrid, Spain; [2]Instituto Ramón y Cajal de Investigación Sanitaria (IRYCIS), Madrid, Spain; [3]Centre National de la Recherche Scientifique (CNRS) and University of Bordeaux, Institut des Maladies Neurodégénératives, Bordeaux, France; [4]Institut National de la Santé et de la Recherche Médicale (INSERM) and University of Bordeaux, NeuroCentre Magendie, Physiopathologie de la Plasticité Neuronale, Bordeaux, France

*For correspondence:
luigi.bellocchio@inserm.fr (LB);
mguzman@quim.ucm.es (MG)

Competing interests: The authors declare that no competing interests exist.

**Abstract** The use of cannabis is rapidly expanding worldwide. Thus, innovative studies aimed to identify, understand and potentially reduce cannabis-evoked harms are warranted. Here, we found that $\Delta^9$-tetrahydrocannabinol, the psychoactive ingredient of cannabis, disrupts autophagy selectively in the striatum, a brain area that controls motor behavior, both in vitro and in vivo. Boosting autophagy, either pharmacologically (with temsirolimus) or by dietary intervention (with trehalose), rescued the $\Delta^9$-tetrahydrocannabinol-induced impairment of motor coordination in mice. The combination of conditional knockout mouse models and viral vector-mediated autophagy-modulating strategies in vivo showed that cannabinoid $CB_1$ receptors located on neurons belonging to the direct (striatonigral) pathway are required for the motor-impairing activity of $\Delta^9$-tetrahydrocannabinol by inhibiting local autophagy. Taken together, these findings identify inhibition of autophagy as an unprecedented mechanistic link between cannabinoids and motor performance, and suggest that activators of autophagy might be considered as potential therapeutic tools to treat specific cannabinoid-evoked behavioral alterations.

## Introduction

Cannabis is one of the most common drugs of abuse in the world (*Alpár et al., 2016*; *Englund et al., 2017*; *Volkow et al., 2014*). Consequently, its major intoxicating constituent, the cannabinoid $\Delta^9$-tetrahydrocannabinol (THC), is the third most popular recreational addictive chemical following ethanol and nicotine. Of note, several states in the USA, as well as a few countries in the world, have legalized the recreational use of cannabis. Cannabis preparations have also been used in medicine for millennia, and nowadays there is a vigorous renaissance in the study and application of their therapeutic effects (*Pertwee, 2012*). In this context, THC and other cannabinoids are already approved by various regulatory agencies, including the Food and Drug Administration (FDA), the European Medicines Agency and Health Canada, as anti-emetic, anti-cachexic, analgesic

and anti-spastic compounds (*Hill, 2015*; *Whiting et al., 2015*). Moreover, medical-grade cannabis dispensation programs have been implemented in about half of the states in the USA and in a growing number of countries globally. However, cannabis use is associated to several undesired and possibly dangerous side effects, so it is crucial that innovative procedures aimed to understand and potentially reduce cannabis-evoked harms are explored (*Alpár et al., 2016*; *Englund et al., 2017*; *Volkow et al., 2014*).

THC exerts its biological effects mainly by activating cannabinoid $CB_1$ receptor, one of the most abundant metabotropic receptors in the mammalian central nervous system (*Katona and Freund, 2008*; *Pertwee et al., 2010*). This receptor is particularly expressed in discrete brain areas involved in the control of learning and memory (cortex, hippocampus), motor behavior (striatum, cerebellum), emotions (amygdala), and autonomic and endocrine functions (hypothalamus, pons, medulla), therefore participating in the control of a wide plethora of biological processes (*Katona and Freund, 2008*; *Mechoulam and Parker, 2013*). A family of retrograde lipid messengers, the endocannabinoids, biologically engages the $CB_1$ receptor, mediating a feedback mechanism aimed to prevent excessive neuronal activity and, thereby, tuning the functionality and plasticity of many synapses (*Castillo et al., 2012*; *Piomelli, 2003*). Recent evidence suggests that the $CB_1$ receptor can control autophagy, a highly conserved and pleiotropic process of cellular 'self-digestion' in which cytoplasmic materials are sequestered into double-membrane vesicles called autophagosomes, and subsequently delivered to lysosomes for degradation or recycling (*Costa et al., 2016*; *Hiebel and Behl, 2015*). Autophagy is an essential mechanism of cellular quality control, and the knowledge on its biological functions in the brain and other organs is rapidly increasing (*Menzies et al., 2017*; *Ohsumi, 2014*). Strikingly, in some cell-culture settings cannabinoids via the $CB_1$ receptor enhance autophagy (*Koay et al., 2014*; *Salazar et al., 2009*), while in others they inhibit autophagy (*Hiebel et al., 2014*; *Piyanova et al., 2013*). Moreover, it is not known yet whether the $CB_1$ receptor controls autophagy in the brain in vivo, and, eventually, what the functional consequences of this potential $CB_1$ receptor/autophagy connection could be. Here, we show that THC inhibits autophagy selectively in the mouse striatum, and that this process participates in the THC-induced impairment of motor coordination. Moreover, administration of clinically safe autophagy activators to mice prevents the dyscoordinating effect of THC. These findings unveil an unprecedented link between cannabinoids, autophagy and motor performance, and provide preclinical evidence for the design of potential new therapeutic strategies aimed at treating specific cannabinoid-induced behavioral alterations.

## Results

### THC impairs striatal autophagy both in vivo and in vitro

To study the effect of THC on autophagy in the brain we first treated wild-type mice with a single i.p. injection of the drug at 10 mg/kg or its vehicle. After 4 hr, we evaluated the status of key autophagy protein markers. This dose and time window allows assessing persistent and pharmacologically tractable behavioral actions of THC administration, as previously reported (*Metna-Laurent et al., 2017*; *Puighermanal et al., 2013*). We analyzed the expression pattern of microtubule-associated light chain three protein (LC3), the most widely used marker of autophagic vesicles (autophagosomes) (*Mizushima et al., 2011*; *Ohsumi, 2014*), in representative brain structures. Upon induction of autophagy, LC3 is converted from a soluble, non-lipidated form (LC3-I) to an aggregated, phosphatidylethanolamine-conjugated form (LC3-II), thereby becoming recruited to autophagosomal membranes (*Mizushima et al., 2011*). THC increased LC3-II levels in the striatum, either when referred to LC3-I (THC vs. vehicle, $t = 4.680$; df = 10; p=0.0009) or to β-actin (THC vs. vehicle, $t = 4.331$; df = 10; p=0.0015) as control, but not in other representative brain regions as the cortex, the hippocampus and the cerebellum (*Figure 1A*). An elevation of LC3-II levels, however, may indicate either that THC increases autophagosome generation (and so increases autophagic flux) or that THC decreases autophagosome clearance (and so decreases autophagic flux) (*Mizushima et al., 2011*; *Ohsumi, 2014*). To discern between these two possibilities, we measured the levels of p62 (sequestosome 1), a pivotal adaptor protein that carries cargo proteins to the autophagosome, being subsequently degraded upon fusion of the autophagosome to the lysosome (autophagolysosome or autolysosome) (*Katsuragi et al., 2015*; *Mizushima et al., 2011*). Hence, an

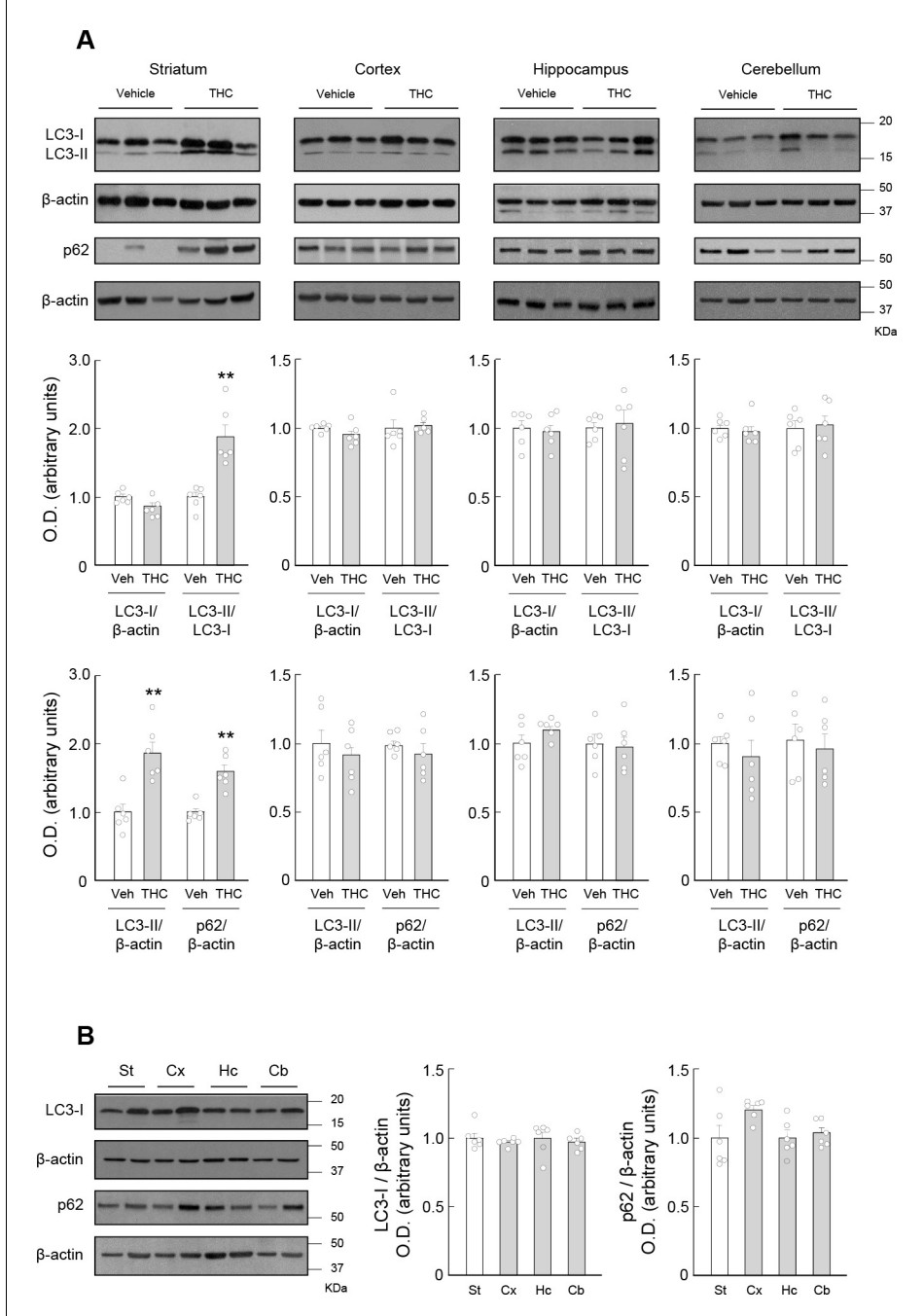

**Figure 1.** THC impairs autophagy in the mouse striatum. Wild-type C57BL/6N mice were treated with THC (10 mg/kg as a single i.p. injection) or its vehicle. Four hours later, the striatum (St), cortex (Cx), hippocampus (Hc) and cerebellum (Cb) were dissected for Western blot analysis. (**A**) Effect of THC on autophagy markers in the different brain regions. (**B**) Relative levels of LC3-I and p62 in the different brain regions from vehicle-treated animals. In both panels, representative blots of each condition, together with optical density values relative to those of the respective loading controls, are shown ($n$ = 6 animals per group). Blots were cropped for clarity. Electrophoretic migration of molecular weight markers is depicted on the right-hand side of each blot. **p<0.01 from vehicle-treated group by unpaired Student $t$-test. Raw numerical data and further statistical details are shown in *Figure 1—source data 1*.

The online version of this article includes the following source data for figure 1:

**Source data 1.** Source data for THC impairs autophagy in the mouse striatum.

increase of LC3-II together with a decrease of p62 usually denotes an active clearance of autophago-somes (and so an increased autophagic flux), while a simultaneous elevation of LC3-II and p62 usually defines an impaired clearance of autophagosomes (and so a decreased autophagic flux) (*Katsuragi et al., 2015*; *Mizushima et al., 2011*). THC induced an accumulation of p62 in the striatum (THC vs. vehicle, $t$ = 5.303; df = 10; p=0.0003), but not in the other brain regions tested (*Figure 1A*), thus indicating that THC impairs the execution of autophagy and that this process occurs selectively in the striatum. Of note, the levels of LC3-I and p62 were not significantly different in the striatum than in the cortex, hippocampus or cerebellum from vehicle-treated mice (*Figure 1B*), suggesting that the selective impact of THC on striatal autophagy does not rely on the basal expression of those two key autophagy proteins but on additional, hitherto unknown molecular factors.

To support a direct action of THC on the striatum, we prepared primary cultures of mouse striatal neurons and treated them with THC (0.75 µM). The synthetic THC analogue HU-210 (10 nM) was used as a further tool to proof pharmacological specificity. The two cannabinoid drugs increased LC3-II levels, as determined by immunofluorescence (aggregated-LC3 *puncta*), in both the total intracellular compartment ($F_{(2,28)}$ = 39.48; THC vs. vehicle, p=0.0004; HU-210 vs. vehicle, p<0.0001) and the lysosomal compartment, as identified by the lysosomal marker LAMP1 ($F_{(2,28)}$ = 22.43; THC vs. vehicle, p<0.0001, HU-210 vs. vehicle: p<0.0001) (*Figure 2A*). Concertedly, they also enhanced p62 levels ($F_{(2,30)}$ = 38.81; THC vs. vehicle, p=0.0130; HU-210 vs. vehicle, p=0.0335) (*Figure 2B*). We next evaluated the effect of the direct inhibition of lysosomal degradation. Upon this downstream blockade of autophagic flux, an autophagy-stimulating compound conceivably induces a further accumulation of autophagosomal markers (LC3-II and p62), while an autophagy-inhibiting compound is not expected to raise those markers further (*Mizushima et al., 2011*). We thus treated striatal neurons with two types of lysosomal inhibitors, specifically the lysosomotropic drug hydroxychloroquine or the lysosomal-protease inhibitors E64d and pepstatin A. As expected, these lysosome-blocking drugs induced per se an accumulation of LC3-II (total LC3-II: $F_{(2,28)}$ = 39.48; hidroxychloroquine vs. vehicle, p<0.0001; E64d/pepstatin A vs. vehicle, p=0.0024. LC3-II/LAMP1: $F_{(2,28)}$ = 22.43; hidroxy-chloroquine vs. vehicle: p<0.0001; E64d/pepstatin A vs. vehicle: p=0.0002) (*Figure 2A*) and p62 ($F_{(2,30)}$ = 38.81; hidroxychloroquine vs. vehicle, p=0.0001; E64d/pepstatin A vs. vehicle, p<0.0001) (*Figure 2B*). Of note, cannabinoids did not significantly heighten those pre-augmented levels of LC3-II (*Figure 2A*) and p62 (*Figure 2B*).

Taken together, these data support that cannabinoids inhibit autophagic flux in striatal neurons both in vivo and in vitro.

## Temsirolimus prevents the THC-induced impairment of striatal autophagy and motor coordination in vivo

THC and other cannabinoids modulate various intracellular signalling pathways in the brain by engaging CB$_1$ receptors (*Castillo et al., 2012*; *Pertwee et al., 2010*). One of the most relevant CB$_1$ receptor-evoked actions is the activation of the phosphatidylinositol-3-kinase/Akt/mammalian target of rapamycin complex 1 (mTORC1) pathway (*Blázquez et al., 2015*; *Gómez del Pulgar et al., 2000*; *Ozaita et al., 2007*; *Puighermanal et al., 2009*). The serine/threonine kinase mTOR, the catalytic component of mTORC1, is critically involved in the control of neural plasticity through the regulation of protein synthesis and other basic cellular functions (*Bockaert and Marin, 2015*; *Lipton and Sahin, 2014*). Of note, mTORC1 is also the most relevant signaling platform that exerts a negative control on autophagy by phosphorylating UNC-51-like kinase 1 (ULK1), and so inhibiting autophagosome formation (*Dunlop and Tee, 2014*; *Saxton and Sabatini, 2017*). However, it is not known whether a cannabinoid-evoked activation of the mTORC1 pathway would be linked to an inhibition of autophagy, and, especially, what the biological consequences of this process could be.

To address this question, we made use of temsirolimus, an FDA-approved rapamycin analogue that selectively blocks mTOR within mTORC1, thereby disinhibiting autophagy (*Dunlop and Tee, 2014*). Mice were treated with temsirolimus (1 mg/kg, i.p.) or its vehicle, and, 20 min later, with THC (10 mg/kg, i.p.) or its vehicle. As THC inhibited autophagy selectively in the striatum, animals were subjected to tests of motor behavior, an archetypical process that - together with, for example, cognition, affection and reward - is controlled by the striatum and is impacted by cannabinoids in both laboratory animals and humans (*Koketsu et al., 2008*; *Kreitzer, 2009*; *Lovinger, 2010*). Four hours after injection, THC impaired motor coordination, as determined by the RotaRod test, and

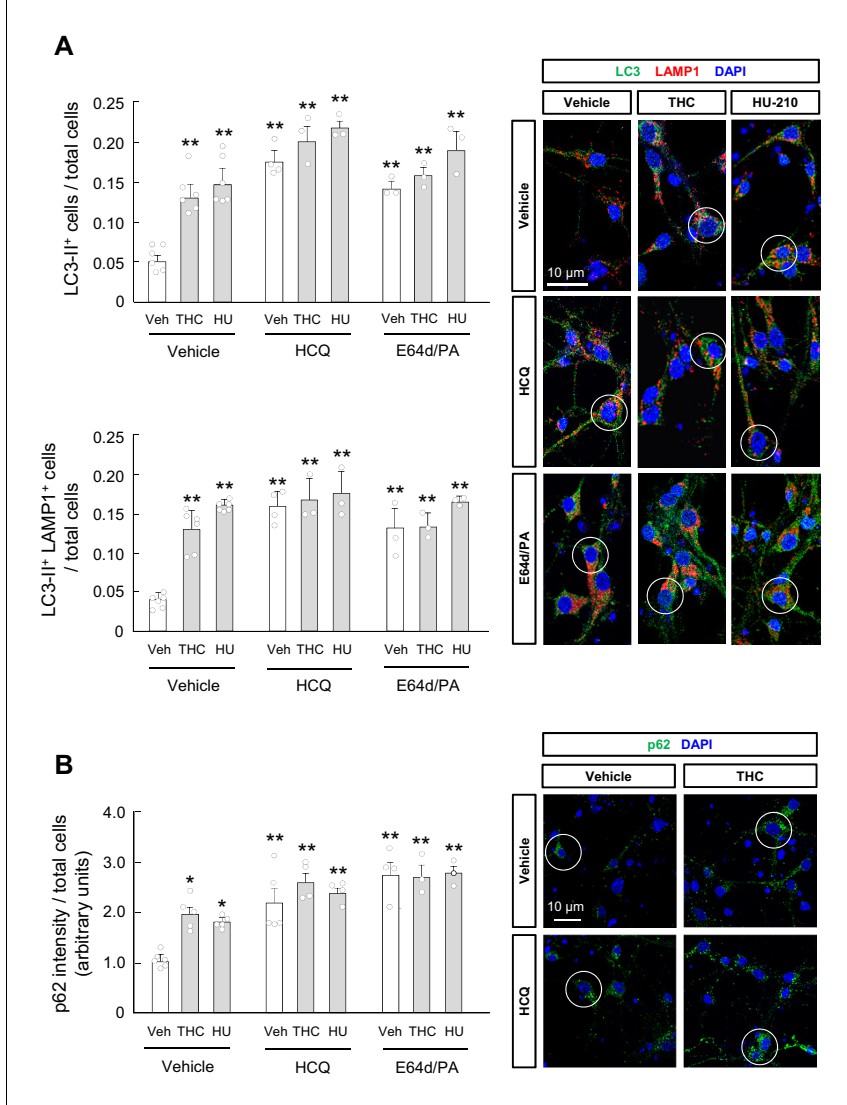

**Figure 2.** THC impairs autophagy in primary striatal neurons. Primary striatal neurons from C57BL/6N mice were exposed for 24 hr to THC (0.75 μM) or HU-210 (10 nM), alone or in combination with hydroxychloroquine (0.1 mM), E64d (0.1 μM) and/or pepstatin A (10 ng/ml), or their vehicles. (**A**) LC3-II immunoreactivity (number of cells with three or more LC3-positive dots relative to total cells; *upper panel*) and LC3-II/LAMP1 immunoreactivity (number of LAMP1-positive cells with three or more LC3 dots relative to total cells; *lower panel*). Representative images with encircled examples of double-positive cells are shown (*n* = 3–6 independent cell preparations per condition). (**B**) p62 immunoreactivity (p62 fluorescence intensity relative to total cells). Representative images of selected experimental conditions with encircled examples of high-intensity cells are shown (*n* = 3–6 independent cell preparations per condition). *p<0.05, **p<0.01 from vehicle-treated group by two-way ANOVA with Tukey's multiple comparisons test. Raw numerical data and further statistical details are shown in *Figure 2—source data 1*.

The online version of this article includes the following source data for figure 2:

**Source data 1.** Source data for THC impairs autophagy in primary striatal neurons.

temsirolimus, under conditions that did not influence behavior by itself, rescued the effect of THC ($F_{(3,30)}$ = 6.635; THC vs. vehicle, p=0.0016; temsirolimus + THC vs. THC, p=0.0096) (*Figure 3A*). In contrast, the inhibitory action of THC on general locomotor activity, as determined by various parameters in the open field test (THC vs. vehicle, ambulation: $F_{(3,24)}$ = 24.14; p=0.0002; activity: $F_{(3,24)}$ = 10.67; p=0.0241; resting time: $F_{(3,24)}$ = 13.89; p=0.0067; fast movements: $F_{(3,24)}$ = 14.15; p=0.0002; stereotypic movements: $F_{(3,24)}$ = 8.240; p=0.0137), was not significantly affected by

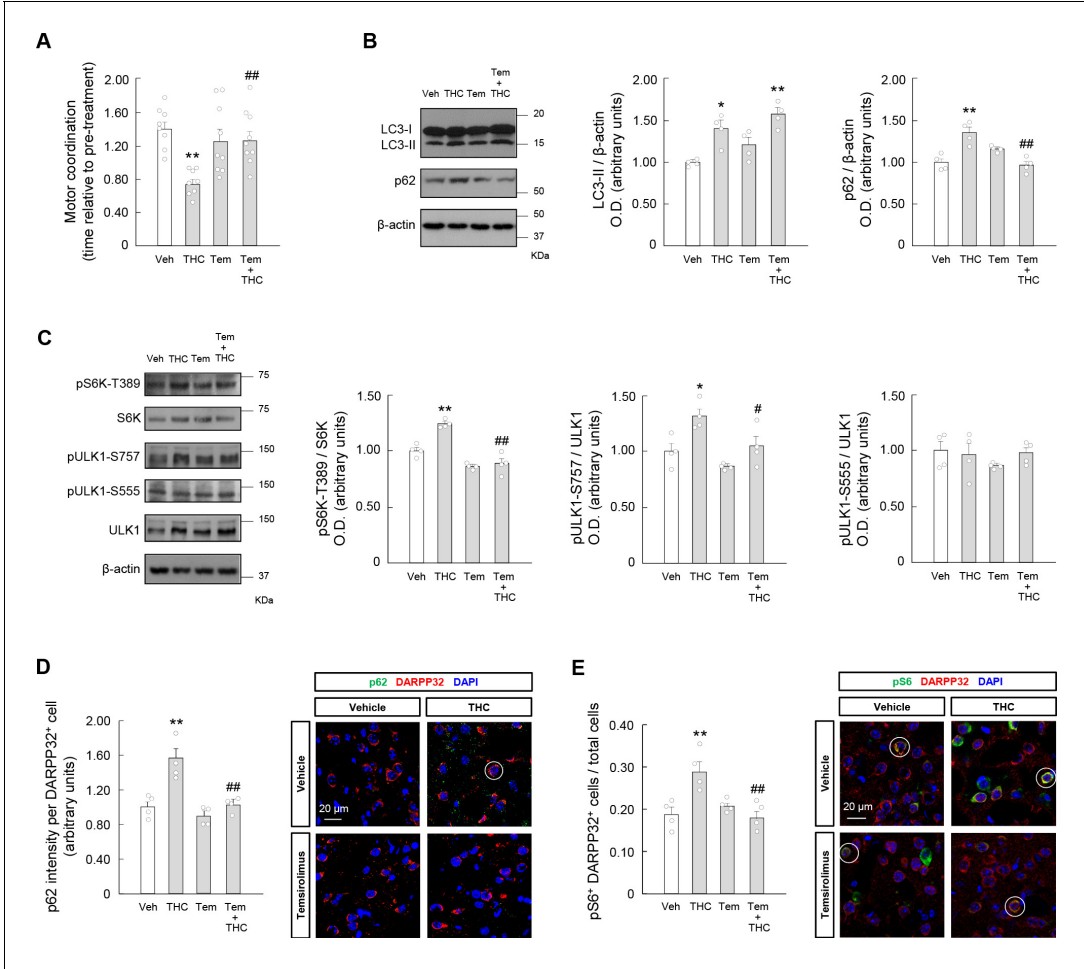

**Figure 3.** Temsirolimus prevents the THC-induced impairment of striatal autophagy and motor coordination in vivo. Wild- type C57BL/6N mice were treated with temsirolimus (1 mg/kg as a single i.p. injection) or its vehicle for 20 min, and, subsequently, with THC (10 mg/kg as a single i.p. injection) or its vehicle for 4 hr. (A) Motor coordination (RotaRod test, time to fall relative to pre-treatment; $n = 8$–9 animals per group). (B, C) Western blot analysis of autophagy markers (*panel B*) and mTORC1 signaling pathway markers (*panel C*) in the striatum. Representative blots of each condition, together with optical density values relative to those of loading controls, are shown ($n = 4$ animals per group). Blots were cropped for clarity. Electrophoretic migration of molecular weight markers is depicted on the right-hand side of each blot. (D, E) Immunofluorescence analysis of p62 (p62 fluorescence intensity per DARPP32-positive cell; *panel D*) and phosphorylated ribosomal protein S6 (phospho-S6/DARPP32 double-positive cells relative to total cells; *panel E*) in the dorsal striatum ($n = 4$ animals per group). Representative images with encircled examples of a high-intensity cell (*panel D*) or double-positive cells (*panel E*) are shown. *p<0.05, **p<0.01 from vehicle-treated group, or #p<0.05, ##p<0.01 from THC-treated group, by one-way ANOVA with Tukey's multiple comparisons test. Raw numerical data and further statistical details are shown in *Figure 3—source data 1*.
The online version of this article includes the following source data and figure supplement(s) for figure 3:

**Source data 1.** Source data for Temsirolimus prevents the THC-induced impairment of striatal autophagy and motor coordination in vivo.
**Figure supplement 1.** Temsirolimus does not rescue THC-induced hypolocomotion.

temsirolimus (*Figure 3—figure supplement 1*; *Puighermanal et al., 2013*). This supports the idea that the THC-induced activation of the mTORC1 pathway selectively affects the coordination component of motor behavior. Under these experimental conditions, western blot analysis of mouse striata supported that THC concomitantly inhibited autophagy, as determined by the simultaneous accumulation of LC3-II (THC vs. vehicle, $F_{(3,12)} = 10.77$; p=0.0119) and p62 (THC vs. vehicle, $F_{(3,12)} = 12.29$; p=0.0017) (*Figure 3B*), and activated the mTORC1 pathway, as determined by an enhanced phosphorylation of the mTORC1-dependent sites in two of its main substrates, namely T389 in 70 kDa ribosomal protein S6 kinase (S6K) (THC vs. vehicle, $F_{(3,12)} = 28.70$; p=0.0009), and S757 in ULK1 (THC vs. vehicle, $F_{(3,12)} = 8.506$; p=0.0214) (*Figure 3C*). Of note, temsirolimus rescued these

cannabinoid-evoked effects on p62 (temsirolimus + THC vs. THC, $F_{(3,12)}$ = 12.29; p=0.0008) (*Figure 3B*) and the mTORC1 pathway markers (pS6K-T389: temsirolimus + THC vs. THC, $F_{(3,12)}$ = 28.70; p<0.0001. pULK1-S757: temsirolimus + THC vs. THC, $F_{(3,12)}$ = 8.506; p=0.0470) (*Figure 3C*). Similar data were obtained by immunofluorescence analysis of p62 levels ($F_{(3,12)}$ = 16.80; THC vs. vehicle, p=0.0009; temsirolimus + THC vs. THC, p=0.0011) (*Figure 3D*), and of the phosphorylation state of the mTORC1/S6K downstream effector ribosomal protein S6 at two target residues (S235/S236) ($F_{(3,12)}$ = 8.34; THC vs. vehicle, p=0.0052; temsirolimus + THC vs. THC, p=0.0047) (*Figure 3E*), in striatal medium spiny neurons (MSNs), as identified by their standard marker dopamine- and cAMP-regulated phosphoprotein of 32 kDa (DARPP32). As a control, we found that the phosphorylation state of the main AMP-activated protein kinase (AMPK)-dependent site in ULK1, namely S555, was not significantly affected by THC and/or temsirolimus (*Figure 3C*). We were unable to immuno-detect significant amounts of LC3 *puncta* in brain sections, which is usually ascribed to the rapid autophagic turnover and very high abundance of LC3-I over LC3-II occurring in living brain tissue (*McMahon et al., 2012*; *Mizushima et al., 2004*; *Sarkar et al., 2014*; *Yang et al., 2011*).

Taken together, these findings suggest that an inhibition of autophagy participates in the motor-dyscoordinating action of THC.

## Trehalose prevents the THC-induced impairment of striatal autophagy and motor coordination in vivo

As a second approach to manipulate autophagy in vivo, we used the natural disaccharide trehalose, which directly stimulates autophagic flux (*Emanuele, 2014*; *Sarkar et al., 2007*). Mice were given trehalose (10 g/L in drinking water) or plain water for 24 hr, and, subsequently, were treated with THC (10 mg/kg, i.p.) or vehicle. Experimental measures were performed 4 hr after acute THC injection. Trehalose, under conditions that did not affect behavior by itself, rescued the THC-evoked impairment of motor coordination ($F_{(3,45)}$ = 3.858; THC vs. vehicle, p=0.0321; trehalose + THC vs. THC, p=0.0358) (*Figure 4A*). As shown above for temsirolimus, the inhibitory action of THC on general locomotor activity, as determined by various parameters in the open field test (THC vs. vehicle, ambulation: $F_{(3,27)}$ = 7.548; p=0.0402; activity: $F_{(3,27)}$ = 8.536; p=0.0134; resting time: $F_{(3,27)}$ = 7.420; p=0.0154; fast movements: $F_{(3,27)}$ = 8.496; p=0.0356; stereotypic movements: $F_{(3,27)}$ = 9.173; p=0.0032), was not significantly affected by trehalose (*Figure 4—figure supplement 1*). Western blot analysis of mouse striata indicated that trehalose reduced the THC-induced accumulation of striatal p62 ($F_{(3,12)}$ = 23.66; THC vs. vehicle, p=0.0017; trehalose + THC vs. THC, p<0.0001) (*Figure 4B*). Trehalose per se did not significantly affect mTORC1 activity markers, but mitigated the THC-evoked stimulation of the pathway (pS6K-T389: $F_{(3,12)}$ = 5.410; THC vs. vehicle, p=0.0388; trehalose + THC vs. THC, p=0.1944. pULK1-S757: $F_{(3,12)}$ = 12.03; THC vs. vehicle, p=0.0032; trehalose + THC vs. THC, p=0.0074) (*Figure 4C*). ULK1-S555 phosphorylation was not significantly affected by THC and/or trehalose (*Figure 4C*). These western blot data were corroborated by immunofluorescence analysis of p62 levels ($F_{(3,12)}$ = 7.575; THC vs. vehicle, p=0.0075; trehalose + THC vs. THC, p=0.0495) (*Figure 4D*) and protein S6 phosphorylation ($F_{(3,12)}$ = 8.822; THC vs. vehicle, p=0.0040; trehalose + THC vs. THC, p=0.0036) (*Figure 4E*).

Taken together, these findings provide further support to the notion that an inhibition of striatal autophagy participates in the motor-dyscoordinating activity of THC.

## Cannabinoid CB₁ receptors located on the direct pathway, but not on cortical projections, are required for the THC-induced impairment of striatal autophagy and motor coordination in vivo

We subsequently studied the neuroanatomical substrate of the observed THC effects. As THC exerts most of its neurobiological effects by activating CB₁ receptors, we first tested the effect of the CB₁ receptor-selective antagonist SR141716 (rimonabant) on the THC-evoked inhibition of motor coordination. Mice were treated with rimonabant (3 mg/kg, i.p.) or vehicle for 20 min, and, subsequently, with THC (10 mg/kg, i.p.) or vehicle. Four hours after injection, THC impaired RotaRod performance, and rimonabant, under conditions that did not influence behavior by itself, abrogated the effect of THC ($F_{(3,16)}$ = 12.86; THC vs. vehicle, p=0.0020; rimonabant + THC vs. THC, p=0.0002) (*Figure 5—figure supplement 1*).

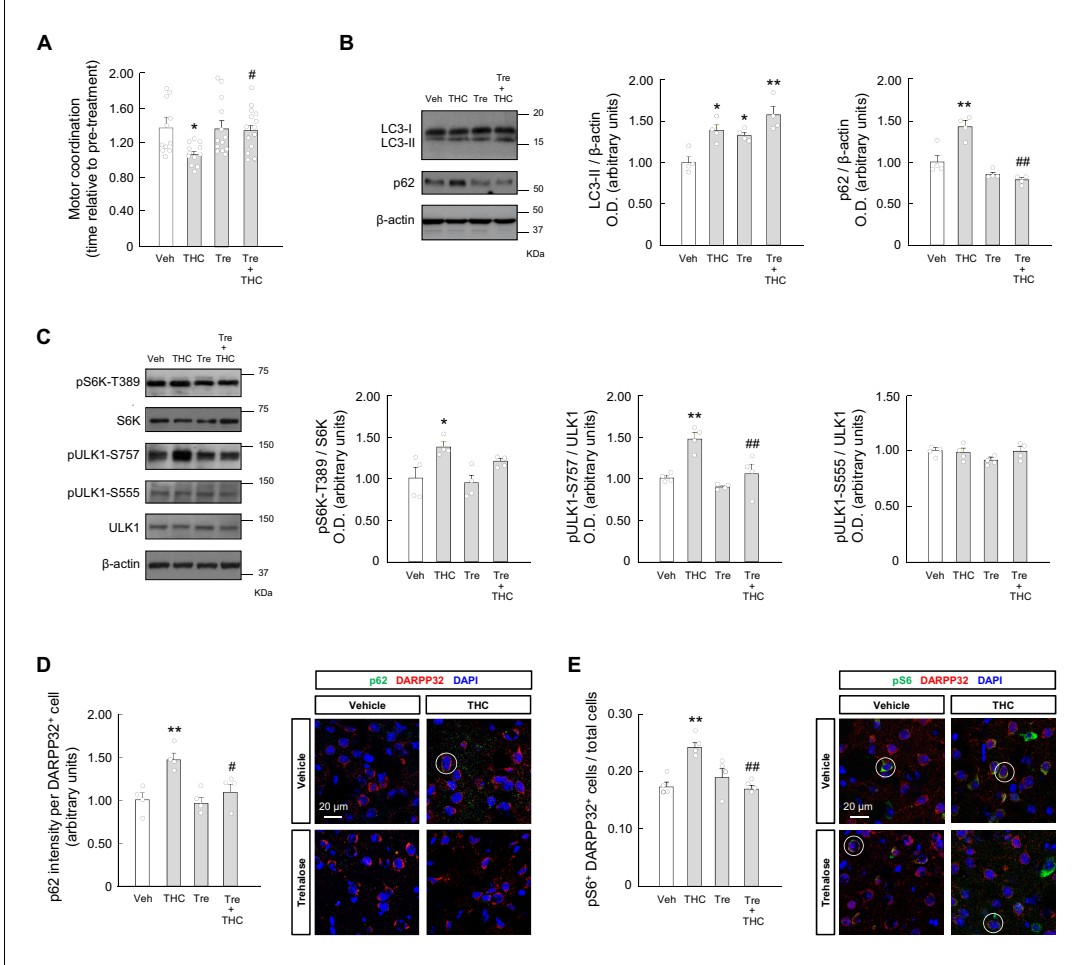

**Figure 4.** Trehalose prevents the THC-induced impairment of striatal autophagy and motor coordination in vivo. Wild-type C57BL/6N mice were given trehalose (10 g/L) or plain water ad libitum for 24 hr, and, subsequently, were treated with THC (10 mg/kg as a single i.p.injection) or its vehicle for 4 hr. (A) Motor coordination (RotaRod test, time to fall relative to pre-treatment; n = 11–14 animals per group). (B, C) Western blot analysis of autophagy markers (*panel B*) and mTORC1 signaling pathway markers (*panel C*) in the striatum. Representative blots of each condition, together with optical density values relative to those of loading controls, are shown (n = 4 animals per group). (D, E) Immunofluorescence analysis of p62 (p62 fluorescence intensity per DARPP32-positive cell; *panel D*) and phosphorylated ribosomal protein S6 (phospho-S6/DARPP32 double-positive cells relative to total cells; *panel E*) in the dorsal striatum (n = 4 animals per group). Representative images with encircled examples of a high-intensity cell (*panel D*) or double-positive cells (*panel E*) are shown. *p<0.05, **p<0.01 from vehicle-treated group, or #p<0.05, ##p<0.01 from THC-treated group, by one-way ANOVA with Tukey's multiple comparisons test. Raw numerical data and further statistical details are shown in *Figure 4—source data 1*.

The online version of this article includes the following source data and figure supplement(s) for figure 4:

**Source data 1.** Source data for Trehalose prevents the THC-induced impairment of striatal autophagy and motor coordination in vivo.

**Figure supplement 1.** Trehalose does not rescue THC-induced hypolocomotion.

Striatal MSNs differ in their neurochemical composition and form two major efferent pathways: the direct (striatonigral) pathway, and the indirect (striatopallidal) pathway (*Kreitzer, 2009*). MSNs in the direct pathway (referred to here as $D_1$R-MSNs) express dopamine $D_1$ receptor ($D_1$R), while MSNs in the indirect pathway (referred to here as $D_2$R-MSNs) express dopamine $D_2$ receptor ($D_2$R). It has been reported that *Cnr1*[fl/fl] mice (referred to here as $CB_1$R-floxed mice) that had been bred with *Drd1a*[Cre] mice to inactivate $CB_1$ receptors selectively in all cells that express $D_1$R (these mice are referred to here as $D_1$R-$CB_1$R KO mice) exhibit a dampened response to the cataleptic effect (but not the overall hypolocomotor effect) of THC (*Monory et al., 2007*). This supports the notion that $CB_1$ receptors located on $D_1$R-MSNs control particular aspects of motor behavior. Hence, we evaluated the RotaRod test in $D_1$R-$CB_1$R KO mice and their control $CB_1$R-floxed littermates. Remarkably, the motor-dyscoordinating action of THC (10 mg/kg, i.p.) found in $CB_1$R-floxed mice was not

evident in $D_1R$-$CB_1R$ KO animals ($F_{(3,18)}$ = 6.571; $CB_1R$-floxed-THC vs. $CB_1R$-floxed-vehicle, p=0.0067; $D_1R$-$CB_1R$ KO-THC vs. $CB_1R$-floxed-THC, p=0.0068) (*Figure 5A*). In concert, the simultaneous accumulation of LC3-II and p62 evoked by THC in control mice, as assessed by western blot, was not observed in $D_1R$-$CB_1R$ KO mice (LC3-II: $F_{(3,12)}$ = 6.981; $CB_1R$-floxed-THC vs. $CB_1R$-floxed-vehicle, p=0.0066; $D_1R$-$CB_1R$ KO-THC vs. $CB_1R$-floxed-THC, p=0.0122. p62: $F_{(3,12)}$ = 13.36; $CB_1R$-floxed-THC vs. $CB_1R$-floxed-vehicle, p=0.0014; $D_1R$-$CB_1R$ KO-THC vs. $CB_1R$-floxed-THC, p=0.0005) (*Figure 5B*). The p62 data were confirmed by immunofluorescence analysis ($F_{(3,9)}$ = 14.36; $CB_1R$-floxed-THC vs. $CB_1R$-floxed-vehicle, p=0.0038; $D_1R$-$CB_1R$ KO-THC vs. $CB_1R$-floxed-THC, p=0.0029) (*Figure 5C*).

By dampening glutamate outflow onto MSNs, $CB_1$ receptors located on corticostriatal projections are considered a key determinant of striatal activity (*Kreitzer, 2009*; *Lovinger, 2010*), mediating, specifically, THC-induced hypolocomotion (*Monory et al., 2007*). We therefore evaluated the possible implication of this $CB_1$ receptor pool in our model. For this purpose, we bred $CB_1R$-floxed mice with *Neurod6*$^{Cre}$ mice to inactivate $CB_1$ receptors selectively in all cells that express NeuroD6 (essentially dorsal telencephalic glutamatergic neurons; these mice are referred to here as Glu-$CB_1R$ KO mice) (*Monory et al., 2006*). Administration of THC (10 mg/kg, i.p.) decreased RotaRod performance comparably in Glu-$CB_1R$ KO mice and their control $CB_1R$-floxed littermates ($F_{(3,12)}$ = 65.18; $CB_1R$-floxed-THC vs. $CB_1R$-floxed-vehicle, p<0.0001; Glu-$CB_1R$ KO-THC vs. Glu-$CB_1R$ KO-vehicle, p<0.0001) (*Figure 5D*). Likewise, as assessed by western blot, THC enhanced LC3-II and p62 levels similarly in Glu-$CB_1R$ KO and $CB_1R$-floxed mice (LC3-II: $F_{(3,12)}$ = 9.471; $CB_1R$-floxed-THC vs. $CB_1R$-floxed-vehicle, p=0.0182; Glu-$CB_1R$ KO-THC vs. Glu-$CB_1R$ KO-vehicle, p=0.0107. p62: $F_{(3,12)}$ = 9.462; $CB_1R$-floxed-THC vs. $CB_1R$-floxed-vehicle, p=0.0093; Glu-$CB_1R$ KO-THC vs. Glu-$CB_1R$ KO-vehicle, p=0.0168) (*Figure 5E*). The p62 data were confirmed by immunofluorescence analysis ($F_{(3,12)}$ = 11.31; $CB_1R$-floxed-THC *vs.* $CB_1R$-floxed-vehicle, p=0.0043; Glu-$CB_1R$ KO-THC *vs.* Glu-$CB_1R$ KO-vehicle, p=0.0108) (*Figure 5F*).

We finally aimed to strengthen the link between the effects of THC on autophagy and motor coordination in $D_1R$-MSNs. We first treated transgenic mice expressing the tdTomato and EGFP reporter genes under the control of the promoter of the *Drd1a* gene (which encodes $D_1R$) and the *Drd2* gene (which encodes $D_2R$), respectively, with THC (10 mg/kg, i.p.) or vehicle. Four hours later, immunofluorescence analysis revealed that the THC-induced activation of the mTORC1 pathway (as determined by protein S6 phosphorylation) occurred selectively in $D_1R$-MSNs ($F_{(3,15)}$ = 7.387; $D_1R$-MSN-THC vs. $D_1R$-MSN-vehicle, p=0.0058; $D_2R$-MSN-THC vs. $D_1R$-MSN-THC, p=0.0146) (*Figure 6—figure supplement 1*). Then, we conducted viral vector-enforced protein expression experiments in vivo. We injected stereotactically into the dorsal striatum of $D_1R$-Cre mice a CAG-DIO rAAV vector carrying a Cre-dependent dominant-negative form of Raptor, one of the essential components of mTORC1 (*Hara et al., 2002*; *Koketsu et al., 2008*), thus allowing the Cre-driven expression of dominant-negative Raptor in $D_1R$-MSNs (c-myc-dnRaptor$^+$ cells: $D_1R$-Cre vs. WT, $t$ = 33.71; df = 6; p<0.0001. pS6$^+$ cells: $F_{(3,9)}$ = 52.52; THC-WT vs. vehicle-WT, p=0.0002; vehicle-$D_1R$-Cre vs. vehicle-WT, p=0.0205; THC-$D_1R$-Cre vs. THC-WT, p<0.0001) (*Figure 6A*). Dominant-negative Raptor did not affect RotaRod performance in vehicle-treated animals, but prevented the THC-induced impairment of motor coordination ($F_{(7,28)}$ = 5.309; THC-WT/post-treatment vs. vehicle-WT/post-treatment, p=0.0010; THC-$D_1R$-Cre/post-treatment vs. THC-WT/post-treatment, p=0.0034) (*Figure 6A*). We next injected stereotactically into the dorsal striatum of $D_1R$-Cre mice a CAG-DIO rAAV vector encoding p62, thus allowing the Cre-driven expression of p62 in $D_1R$-MSNs (p62 intensity: $D_1R$-Cre vs. WT, $t$ = 12.22; df = 6; p<0.0001) (*Figure 6B*). Of note, p62 overexpression per se decreased RotaRod performance in vehicle-treated animals, and this decrease was not additive to that induced by THC administration (10 mg/kg, i.p.) ($F_{(7,28)}$ = 5.641; THC-WT/post-treatment vs. vehicle-WT/post-treatment, p=0.0152; vehicle-$D_1R$-Cre/pre-treatment vs. vehicle-WT/pre-treatment, p=0.0477; THC-$D_1R$-Cre/pre-treatment vs. THC-WT/pre-treatment, p=0.0450) (*Figure 6B*).

Taken together, all these findings indicate that $CB_1$ receptors located on $D_1R$-MSNs, but not on corticostriatal projections, are required for the autophagy-inhibiting and motor-dyscoordinating activity of THC.

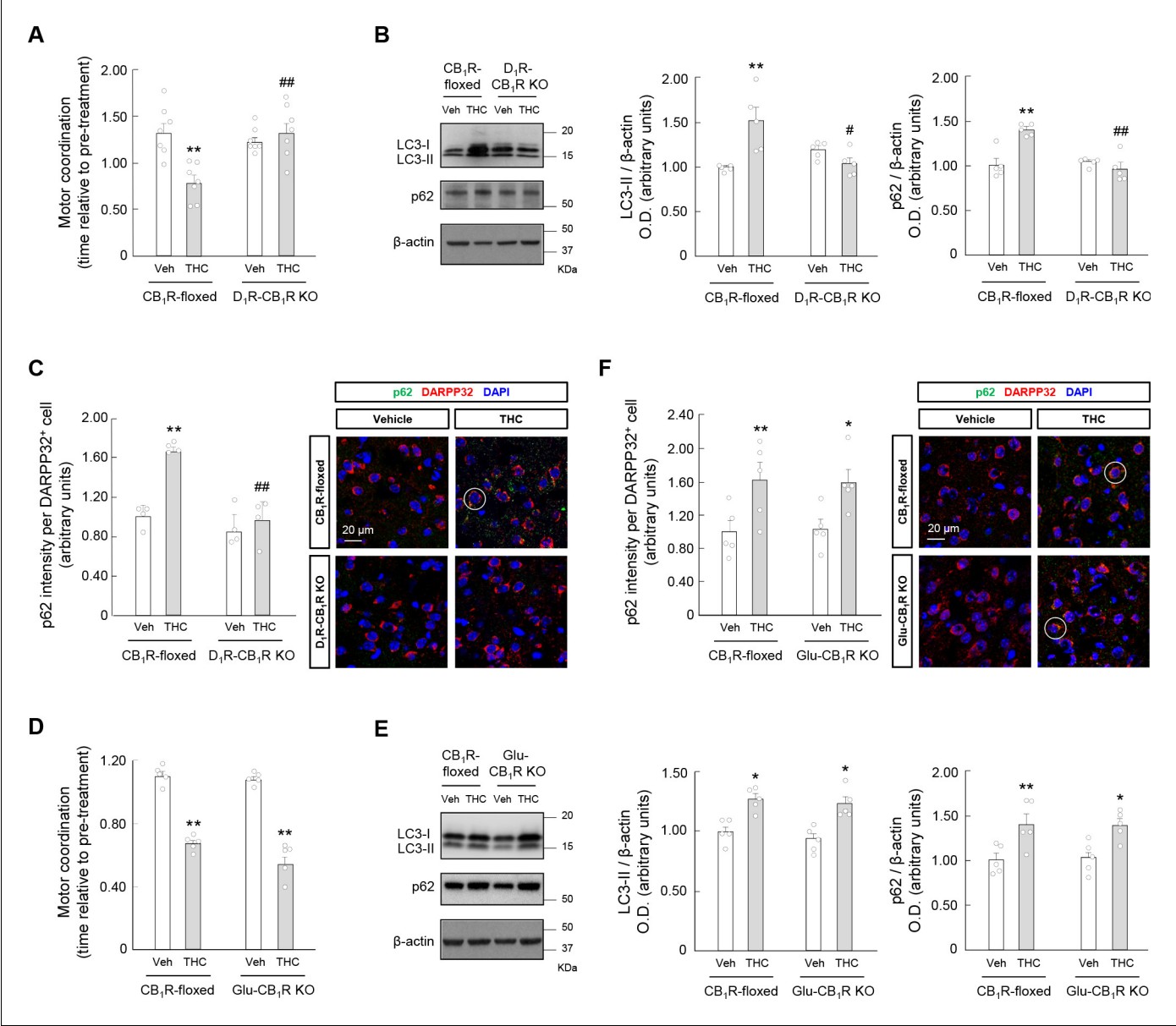

**Figure 5.** Cannabinoid CB$_1$ receptors located on D$_1$R-MSNs, but not on glutamatergic neurons, are required for the THC-induced impairment of striatal autophagy and motor coordination in vivo. (**A–C**) D$_1$R-CB$_1$R KO mice and CB$_1$R-floxed control littermates were treated with THC (10 mg/kg as a single i. p.injection) or its vehicle for 4 hr. *Panel A*, Motor coordination (RotaRod test, time to fall relative to pre-treatment; n = 7 animals per group). *Panel B*, Western blot analysis of autophagy markers in the striatum. Representative blots of each condition, together with optical density values relative to those of loading controls, are shown (n = 5 animals per group). Blots were cropped for clarity. Electrophoretic migration of molecular weight markers is depicted on the right-hand side of each blot. *Panel C*, Immunofluorescence analysis of p62 (p62 fluorescence intensity per DARPP32-positive cell) in the dorsal striatum (n = 4 animals per group). Representative images with an encircled example of high-intensity cell are shown. (**D–F**) Glu-CB$_1$R KO mice and CB$_1$R-floxed control littermates were treated with THC (10 mg/kg as a single i.p. injection) or its vehicle for 4 hr. *Panel D*, Motor coordination (RotaRod test, time to fall relative to pre-treatment; n = 5 animals per group). *Panel E*, Western blot analysis of autophagy markers in the striatum. Representative blots of each condition, together with optical density values relative to those of loading controls, are shown (n = 5 animals per group). Blots were cropped for clarity. Electrophoretic migration of molecular weight markers is depicted on the right-hand side of each blot. *Panel F*, Immunofluorescence analysis of p62 (p62 fluorescence intensity per DARPP32-positive cell) in the dorsal striatum (n = 4 animals per group). Representative images with an encircled example of high-intensity cell are shown. *p<0.05, **p<0.01 from the corresponding vehicle-treated group, or #p<0.05, ##p<0.01 from the corresponding THC-treated CB$_1$R-floxed group, by two-way ANOVA with Tukey's multiple comparisons test. Raw numerical data and further statistical details are shown in *Figure 5—source data 1*.

The online version of this article includes the following source data and figure supplement(s) for figure 5:

*Figure 5 continued on next page*

*Figure 5 continued*

**Source data 1.** Source data for Cannabinoid $CB_1$ receptors located on $D_1R$-MSNs, but not on glutamatergic neurons, are required for the THC-induced impairment of striatal autophagy and motor coordination in vivo.

**Figure supplement 1.** Rimonabant rescues THC-induced motor dyscoordination.

## Discussion

Here, we identify impairment of autophagy as an unprecedented mechanism involved in cannabinoid-induced motor alterations. On molecular grounds, our data favour a 'two-hit' model by which engagement of striatal $CB_1$ receptors may impair autophagy. First, $CB_1$ receptor activation, by coupling to the phosphatidylinositol-3-kinase/Akt/mTORC1 pathway, would lead to ULK1 phosphorylation, which, subsequently, would inhibit autophagosome formation/autophagy initiation. Second, $CB_1$ receptor activation, by a hitherto undefined mechanism that may conceivably involve an impact on lysosomal function (*Hiebel and Behl, 2015*), would inhibit autophagosome clearance/autophagy completion. We are aware, however, that our work has several shortcomings that could limit the generalization of its conclusions. Specifically, (*i*) the data (except for the cell-culture experiments) come from a single cannabinoid agonist (THC) given at a single dose (10 mg/kg, i.p.), and were collected at a single time point after administration (4 hr); (*ii*) only two (albeit well-established) motor behavior measures were examined (RotaRod and open field); and (*iii*) only male animals were studied.

By targeting mTORC1 with temsirolimus, we report a feasible pharmacological intervention to rescue the concerted THC-evoked impairment of autophagy and motor coordination. Temsirolimus prevents other unwanted effects of THC, such as short-term memory loss and anxiety, leaving potential therapeutically sought cannabinoid actions as analgesia and anxiolysis unaffected (*Puighermanal et al., 2013*). Temsirolimus has similar potency and specificity for mTOR than rapamycin, but longer stability and increased solubility, and is already approved by the FDA as first-line treatment for metastatic renal cancer patients classified as poor risk (*Hudes et al., 2007*). In these patients, temsirolimus is well-tolerated, increases overall survival, and improves quality of life (*Zanardi et al., 2015*). Taken together, these pieces of evidence suggest that administration of temsirolimus, or other FDA-approved rapalogs like everolimus (*Janku et al., 2018*; *Lebwohl et al., 2013*), might help to counteract some particular unwanted effects of cannabis.

Dietary manipulation with trehalose also prevented the THC-evoked impairment of autophagy and motor dyscoordination. Trehalose, a nontoxic disaccharide found in numerous plants, microorganisms and invertebrates, contains an $\alpha,\alpha-1,1$-glucoside bond between two $\alpha$-glucose units, thus being an extremely stable sugar. In many countries, including USA, trehalose is added to various food products as nutritional supplement and 'natural flavor' (*Richards et al., 2002*). On physiological grounds, trehalose is believed to stabilize proteins and to protect them from stress-induced unfolding, aggregation and degradation (*Emanuele, 2014*; *Hosseinpour-Moghaddam et al., 2018*). Vertebrates cannot synthesize trehalose, but exogenous trehalose administration induces the clearance of toxic protein aggregates in cultured mammalian cells, and exerts therapeutic effects in a plethora of mouse models of protein-misfolding disorders (including Huntington's disease, Alzheimer's disease, Parkinson's disease and amyotrophic lateral sclerosis) concomitantly to autophagy induction (*Hosseinpour-Moghaddam et al., 2018*; *Menzies et al., 2017*). Although its mechanism of action is not completely understood (*Lee et al., 2018*), trehalose has been proposed to activate autophagy via competitive inhibition of GLUT glucose transporters, thus impairing cellular energy supply and stimulating AMPK (*DeBosch et al., 2016*). In our hands, however, the phosphorylation state of the main AMPK-dependent site in ULK1 remained unaffected upon trehalose treatment. We also observed that, in line with some reports (e.g. *Sarkar et al., 2007*), trehalose did not affect the basal activity of mTORC1-pathway molecular markers; and, in line with other reports (e.g. *DeBosch et al., 2016*), it attenuated stimulus-evoked mTORC1 overactivation. Thus, it is likely that AMPK, mTORC1, and the contextual crosstalk between these two pivotal signalling axes (*Alers et al., 2012*) are required for the full pro-autophagic effects of trehalose to be observed.

We also define here the neuroanatomical basis for the autophagy-inhibiting and motor-dyscoordinating actions of THC. The $CB_1$ receptor is one of the most abundant metabotropic receptors in the striatum, where it is mainly expressed in $D_1R$-MSNs, $D_2R$-MSNs, GABAergic interneurons, and

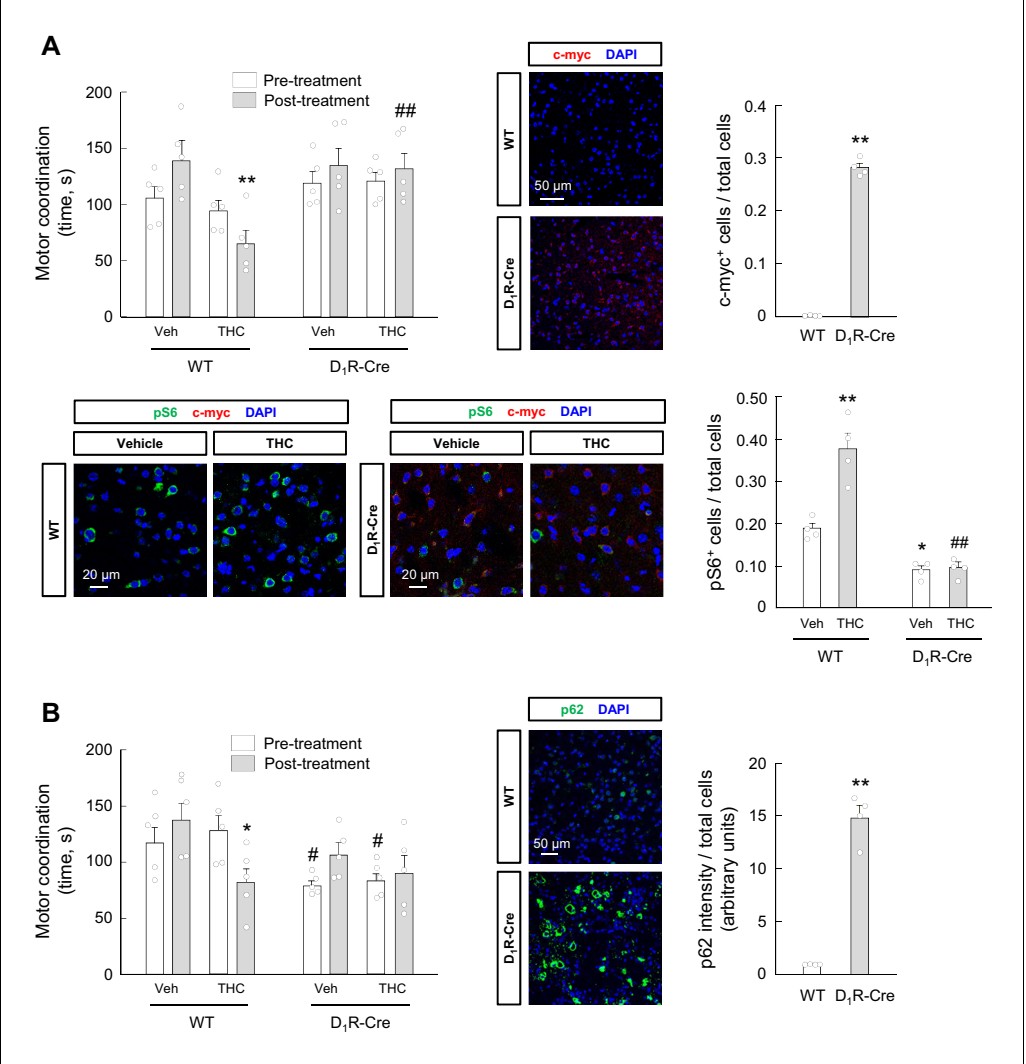

**Figure 6.** mTORC1 and p62 in $D_1$R-MSNs participate in the THC-induced impairment of motor coordination in vivo. (**A**) $D_1$R-Cre mice and wild-type control littermates were injected stereotactically into the dorsal striatum with a CAG-DIO-dnRaptor rAAV, and left untreated for 4 weeks. Animals were subsequently treated with THC (10 mg/kg as a single i.p. injection) or its vehicle for 4 hr, and motor coordination was evaluated (RotaRod test, time to fall in seconds; $n$ = 5 animals per group). **$p<0.01$ from vehicle-treated WT/post-treatment group, or ##$p<0.01$ from THC-treated WT/post-treatment group, by two-way ANOVA with Tukey's multiple comparisons test. Representative images of c-myc tag and phosphorylated ribosomal protein S6 staining in the dorsal striatum, together with their quantification (c-myc-positive cells relative to total cells, or phospho-S6-positive cells relative to total cells), are shown ($n$ = 4 animals per group). **$p<0.01$ from WT group by unpaired Student $t$-test (*c-myc immunofluorescence*); *$p<0.05$, **$p<0.01$ from vehicle-treated/WT group, or ##$p<0.01$ from THC-treated/WT group, by two-way ANOVA with Tukey's multiple comparisons test (*phospho-S6 immunofluorescence*). (**B**) $D_1$R-Cre mice and wild-type control littermates were injected stereotactically into the dorsal striatum with a CAG-DIO-p62 rAAV, and left untreated for 4 weeks. Animals were subsequently treated with THC (10 mg/kg as a single i.p. injection) or its vehicle for 4 hr, and motor coordination was evaluated (RotaRod test, time to fall in seconds; $n$ = 5 animals per group). *$p<0.05$ from vehicle-treated WT/post-treatment group, or #$p<0.05$ from the respective WT/pre-treatment group, by two-way ANOVA with Tukey's multiple comparisons test. Representative images of p62 staining in the dorsal striatum, together with their quantification (p62 fluorescence intensity relative to total cells), are shown ($n$ = 4 animals per group). **$p<0.01$ from WT group by unpaired Student $t$-test. Raw numerical data and further statistical details are shown in *Figure 6—source data 1*.

The online version of this article includes the following source data and figure supplement(s) for figure 6:

**Source data 1.** Source data for mTORC1 and p62 in $D_1$R-MSNs participate in the THC-induced impairment of motor coordination in vivo.

*Figure 6 continued on next page*

*Figure 6 continued*

**Figure supplement 1.** THC activates the mTORC1 pathway in D$_1$R-MSNs but not D$_2$R-MSNs in vivo.

astrocytes, as well as in glutamatergic terminals projecting from the cortex (*Castillo et al., 2012*; *Davis et al., 2018*; *Uchigashima et al., 2007*). This complex anatomical profile dictates an intricate repertoire of modulatory actions controlled by endocannabinoids through different CB$_1$ receptor pools, ranging from synaptic plasticity (*Covey et al., 2017*; *Kreitzer, 2009*) to astrocyte-neuron communication (*Araque et al., 2017*) and neuronal integrity (*Chiarlone et al., 2014*; *Naydenov et al., 2014*). Specifically, our data show that the pool of CB$_1$ receptors located on D$_1$R-MSNs plays an indispensable role in cannabinoid-induced impairment of autophagy and motor coordination. Of note, it has been shown that this precise CB$_1$ receptor subpopulation is also necessary for cannabinoid-induced catalepsy, although not for overall cannabinoid-induced hypomotility (*Monory et al., 2007*), thus supporting that, in agreement with our data, it controls selected aspects of motor behavior. In addition, CB$_1$ receptors located on corticostriatal terminals, by controlling glutamatergic signalling, contribute to THC-induced hypolocomotion (*Monory et al., 2007*), participate in endocannabinoid-dependent long-term depression as evoked by D$_1$R-MSNs (*Bagetta et al., 2011*; *Wu et al., 2015*) and protect D$_1$R-MSNs from toxic insults (*Ruiz-Calvo et al., 2018*). Thus, endocannabinoid signalling fine-tunes the functions and viability of D$_1$R-MSNs through a delicate armamentarium of CB$_1$ receptor pools located on both D$_1$R-MSNs, presynaptic terminals impinging on them, and other surrounding cell types.

We note that our work does not unveil the precise cellular and molecular mechanisms by which the CB$_1$ receptor-evoked inhibition of autophagy in D$_1$R-MSNs affects brain functionality to change motor coordination. Neuronal communication is finely sensitive to proteostatic processes as autophagy, which, for example, clears dysfunctional proteins and fine-tunes the trafficking/recycling of membrane neurotransmitter receptors (e.g. ionotropic glutamate receptors; *Birdsall and Waites, 2019*). Neuronal activity is associated to the mTORC1 pathway and autophagy, and this could in turn participate in NMDA receptor-dependent synaptic plasticity and brain function (*Shehata et al., 2012*). Hence, the control of long-term depression exerted by CB$_1$ receptors on D$_1$R-MSNs (*Bagetta et al., 2011*; *Wu et al., 2015*) might be mechanistically connected to the THC-evoked effects on mTORC1/autophagy reported here, perhaps by signaling in a cell-autonomous manner through the accumulation of the multifunctional scaffold protein p62 (*Sánchez-Martín and Komatsu, 2018*). These possibilities notwithstanding, our findings might also be relevant in other neurobiological processes that are known to be controlled by the striatum and impacted by cannabinoids - for example cognition, affection and reward (*Katona and Freund, 2008*; *Kreitzer, 2009*; *Lovinger, 2010*). Moreover, from a translational point of view, it is tempting to speculate that D$_1$R-MSNs, but not corticostriatal terminals, would constitute the neuroanatomical target of strategies aimed at managing some specific cannabis-induced behavioral alterations as catalepsy and dyscoordination.

# Materials and methods

## Key resources table

| Reagent type (species) or resource | Designation | Source or reference | Identifiers | Additional information |
|---|---|---|---|---|
| Strain, strain background (*Mus musculus*, C57BL/6N, male) | *Cnr1$^{fl/fl}$;Drd1a$^{Cre}$* | *Monory et al., 2007*; doi:10.1371/journal.pbio.0050269 | N/A | Conditional mutant mice in which the CB$_1$ receptor gene (*Cnr1*) is absent from D$_1$R (*Drd1a*)-expressing cells |
| Strain, strain background (*Mus musculus*, C57BL/6N, male) | *Drd1a$^{Cre}$* | *Lemberger et al., 2007*; doi:10.1186/1471-2202-8-4 | N/A | Transgenic mice expressing Cre recombinase in D$_1$R (*Drd1a*)-expressing cells |

*Continued on next page*

*Continued*

| Reagent type (species) or resource | Designation | Source or reference | Identifiers | Additional information |
|---|---|---|---|---|
| Strain, strain background (*Mus musculus*, C57BL/6N, male) | *Cnr1*$^{fl/fl}$;*Neurod6*$^{Cre}$ | *Monory et al., 2006*; doi:10.1016/j.neuron.2006.07.006 | N/A | Conditional mutant mice in which the CB$_1$ receptor gene (*Cnr1*) is absent from dorsal telencephalic glutamatergic (*Neurod6*-expressing) neurons |
| Strain, strain background (*Mus musculus*, C57BL/6N, male) | *Drd1a-tdTomato*; *Drd2-EGFP* | *Suárez et al., 2014*; doi:10.1016/j.biopsych.2013.05.006 | N/A | Transgenic mice expressing the tdTomato and EGFP reporter genes under the control of the *Drd1a* gene promoter and the *Drd2* gene promoter, respectively |
| Strain, strain background (*Mus musculus*, C57BL/6N, male) | C57BL/6N | Harlan Laboratories | RRID:MGI:5902763 | Wild-type mice |
| Transfected construct (*Homo sapiens*) | Myc-Raptor (ΔCT) expression vector | Addgene *Hara et al., 2002*; doi:10.1016/s0092-8674(02)00833–4. *Koketsu et al., 2008*; doi:10.1152/ajpendo.00253.2007 | Plasmid #1859; RRID:Addgene_1859 | Vector backbone: pRK-5; construct generated by PCR-mediated deletion of 1293 base pairs at the Raptor *C*-terminus |
| Transfected construct (*Homo sapiens*) | HA-p62 expression vector | Addgene | Plasmid #28027; RRID:Addgene_28027 | Vector backbone: pcDNA4/TO |
| Genetic reagent (*Homo sapiens*) | CAG-DIO rAAV expression vector | *Klugmann et al., 2005*; doi:10.1016/j.mcn.2004.10.002 *Bellocchio et al., 2016*; doi:10.1523/JNEUROSCI.1192–16.2016 | CAG-DIO rAAV Hybrid serotype 1/2 | Recombinant adeno-associated virus (rAAV) for Cre-driven transgene expression with a CAG promoter |
| Biological sample (*Mus musculus*) | Primary striatal neurons | Harlan Laboratories (C57BL/6N mice) | C57BL/6N RRID:MGI:5902763 | In vitro cell cultures |
| Antibody | Anti-LC3B (rabbit polyclonal) | Sigma-Aldrich | Cat. #L7543; RRID:AB_796155 | IF (1:300); WB (1:4000) |
| Antibody | Anti-p62 (rabbit polyclonal) | Enzo Life Sciences | Cat. #BML-PW9860-0025; RRID:AB_2052149 | IF (1:250); WB (1:1000) |
| Antibody | Anti-p62 (rabbit polyclonal) | Progen | Cat. #GP62-C; RRID:AB_2687531 | WB (1:1000) |
| Antibody | Anti-LAMP1 (rabbit polyclonal) | Abcam | Cat. #ab25245 RRID:AB_449893 | IF (1:1000) |
| Antibody | Anti-DARPP32 (mouse monoclonal) | BD Biosciences | Cat. #611520; RRID:AB_398980 | IF (1:700) |
| Antibody | Anti-phospho-S6-S235/S236 (rabbit polyclonal) | Cell Signaling | Cat. #2211; RRID:AB_331679 | IF (1:300) |
| Antibody | Anti-phospho-S6-S240/S244 (rabbit polyclonal) | Cell Signaling | Cat. #5364; RRID:AB_10694233 | IF (1:800) |
| Antibody | Anti-c-Myc (mouse monoclonal) | Sigma-Aldrich | Cat. #11-667-149-001; RRID:AB_390912 | IF (1:500) |
| Antibody | Anti-phospho-S6K-T389 (mouse monoclonal) | Cell Signaling | Cat. #9206; RRID:AB_2285392 | WB (1:1000) |
| Antibody | Anti-total-S6K (rabbit polyclonal) | Cell Signaling | Cat. #9202; RRID:AB_331676 | WB (1:1000) |

*Continued on next page*

*Continued*

| Reagent type (species) or resource | Designation | Source or reference | Identifiers | Additional information |
|---|---|---|---|---|
| Antibody | Anti-phospho-ULK1-S757 (rabbit polyclonal) | Cell Signaling | Cat. #6888; RRID:AB_10829226 | WB (1:1000) |
| Antibody | Anti-phospho-ULK1-S555 (rabbit polyclonal) | Cell Signaling | Cat. #5869; RRID:AB_10707365 | WB (1:1000) |
| Antibody | Anti-total-ULK1 (rabbit polyclonal) | Cell Signaling | Cat. #8054; RRID:AB_11178668 | WB (1:1000) |
| Antibody | Anti-β-actin (mouse monoclonal) | Sigma-Aldrich | Cat. #A5441; RRID:AB_476744 | WB (1:4000) |
| Antibody | Anti-mouse monoclonal IgG (HRP-linked whole antibody) | GE-Healthcare Lifescience | Cat. #NA931; RRID:AB_772210 | WB (1:5000) |
| Antibody | Anti-rabbit monoclonal IgG (HRP-linked whole antibody) | GE-Healthcare Lifescience | Cat. #NA934; RRID:AB_2722659 | WB (1:5000) |
| Antibody | Goat anti-guinea pig IgG (H+L) (HRP-linked secondary antibody) | Invitrogen | Cat. #A18769; RRID:AB_2535546 | WB (1:5000) |
| Antibody | Goat anti-mouse IgG (H+L) (cross-adsorbed, Alexa Fluor 488) | Invitrogen | Cat. #A-11001; RRID:AB_2534069 | IF (1:500) |
| Antibody | Goat anti-mouse IgG (H+L) (cross-adsorbed, Alexa Fluor 594) | Invitrogen | Cat. #A-11005; RRID:AB_2534073 | IF (1:500) |
| Antibody | Goat anti-mouse IgG (H+L) (cross-adsorbed, Alexa Fluor 647) | Invitrogen | Cat. #A-21235; RRID:AB_2535804 | IF (1:500) |
| Antibody | Goat anti-rabbit IgG (H+L) (cross adsorbed, Alexa Fluor 488) | Invitrogen | Cat. #A-11008; RRID:AB_143165 | IF (1:500) |
| Antibody | Goat anti-rabbit IgG (H+L) (cross adsorbed, Alexa Fluor 594) | Invitrogen | Cat. #A-11012; RRID:AB_2534079 | IF (1:500) |
| Antibody | Goat anti-rabbit IgG (H+L) (cross adsorbed, Alexa Fluor 647) | Invitrogen | Cat. #A-21244; RRID:AB_2535812 | IF (1:500) |
| Commercial assay or kit | Papain dissociation system (PDS) | Worthington | Cat. #LK 003153 | In vitro cell cultures |
| Chemical compound, drug | $\Delta^9$-tetrahydro-cannabinol (THC) | THC Pharm GmbH | Dronabinol | In vivo experiments (10 mg/kg, i.p.); in vitro experiments (0.75 µM) |
| Chemical compound, drug | Rimonabant (SR141716) | Cayman Chemical | Cat. #9000484 | In vivo experiments: (3 mg/kg, i.p.) |
| Chemical compound, drug | Temsirolimus | LC Labs | Cat. #T-8040 | In vivo experiments (1 mg/kg, i.p.) |

*Continued on next page*

*Continued*

| Reagent type (species) or resource | Designation | Source or reference | Identifiers | Additional information |
|---|---|---|---|---|
| Chemical compound, drug | Trehalose | Merck-Calbiochem | Cat. #90210 | In vivo experiments (10 g/L in drinking water) |
| Chemical compound, drug | HU-210 | Tocris | Cat. #0966 | In vitro experiments (10 nM) |
| Chemical compound, drug | Hydroxychloroquine | Merck | Cat. #509272 | In vitro experiments (0.1 mM) |
| Chemical compound, drug | E64d | Enzo Life Sciences | Cat. #BML-PI107-0001 | In vitro experiments (0.1 μM) |
| Chemical compound, drug | Pepstatin A | Enzo Life Sciences | Cat. #ALX-260–085 M005 | In vitro experiments (10 ng/ml) |
| Software, algorithm | Graph Pad Prism 8.0 | GraphPad Software Inc | RRID:SCR_002798 | Descriptive analysis and statistics |
| Software, algorithm | IBM SPSS | IBM Corporation | RRID:SCR_002865 | Statistical power analysis |
| Software, algorithm | Image J | NIH | RRID:SCR_003070 | Western blot and immune-microscopy image analysis |
| Software, algorithm | TCS-SP8 Leica Application Suite X, LASX | Leica | RRID:SCR_013673 | SP8 AOBS confocal microscopy image capture |
| Software, algorithm | ACTITRACKUPG V2.7 | Panlab | Cat. #76–0610 | Motor activity analysis |
| Other | DAPI stain | Invitrogen | Cat. #D1306; RRID:AB_2629482 | IF (1 μg/mL) |
| Other | RotaRod LE8200 | Harvard Apparatus | Cat. #LE8200 (76–0237) | Motor coordination testing |
| Other | IR actimeter (ActiTrack) | Panlab | Cat. #76–0127, #76–0131, #76–0134, #76–0125 | Motor activity testing |

## Animals

We used conditional mutant mice, generated by the Cre-loxP technology, in which the $CB_1$ receptor gene (*Cnr1*) is absent either from $D_1R$-expressing neurons (*Cnr1$^{fl/fl}$* mice bred with *Drd1a$^{Cre}$* mice; referred to here as $D_1R$-$CB_1R$ KO mice) (*Monory et al., 2007*) or from dorsal telencephalic glutamatergic neurons (*Cnr1$^{fl/fl}$* mice bred with *Neurod6$^{Cre}$* mice; referred to here as Glu-$CB_1R$ KO mice) (*Monory et al., 2006*), as well as their respective *Cnr1$^{fl/fl}$* (referred to here as $CB_1R$-floxed) littermates. We also used BAC transgenic mice expressing the tdTomato and EGFP reporter genes under the control of the *Drd1a* gene promoter and *Drd2* gene promoter, respectively (*Drd1a-tdTomato; Drd2-EGFP* mice; colony founders kindly provided by Rosario Moratalla, Cajal Institute, Madrid, Spain) (*Suárez et al., 2014*). Wild-type C57BL/6N mice were purchased from Harlan Laboratories (Barcelona, Spain). Animal housing, handling and assignment to the different experimental groups were conducted essentially as described before (*Bagetta et al., 2011*). Throughout the study, animals had unrestricted access to food and water. They were housed (4–5 mice per cage) under controlled temperature (range, 20–22°C), humidity (range, 50–55%) and light/dark cycle (light between 8:00 am and 8:00 pm). Animals were habituated to housing conditions before the start of the experiments, were assigned randomly to the different treatment groups, and all experiments were performed in a blinded manner for genotype, pharmacological treatment and viral injection. All animals used in the experiments were male adults (*ca.* 8-week-old). Adequate measures were taken to minimize pain and discomfort of the animals, as well as the number of animals used in the experiments, on the basis of the 3Rs (replacement, reduction and refinement) principle. Mice were sacrificed either by intracardial perfusion with paraformaldehyde (and their brains subsequently excised for histological analyses) or by cervical dislocation (and their striata, or other brain regions, subsequently dissected for Western blot analyses). All the experimental procedures were performed in accordance with the guidelines and with the approval of the Animal Welfare Committee of Universidad

Complutense de Madrid and Comunidad de Madrid (PROEX 209/18), and in accordance with the directives of the European Commission (2010/63/EU).

## Drug treatments in vivo

THC (THC Pharm GmbH, Frankfurt am Main, Germany) and SR141716 (rimonabant; Cayman Chemical, Ann Arbor, MI, USA) were stored in DMSO. Just before the experiments, solutions of vehicle [1% (v/v) DMSO in Tween-80/saline (1:80, v/v)], THC (10 mg/kg body weight) and/or rimonabant (3 mg/kg body weight) were prepared for i.p. injections. When rimonabant was used (*Figure 5—figure supplement 1*), mice were treated with rimonabant (3 mg/kg as a single i.p. injection) or vehicle for 20 min, and, subsequently, with THC (10 mg/kg as a single i.p. injection) or vehicle for 4 hr. Temsirolimus (LC Labs, Woburn, MA, USA) was prepared fresh in DMSO just before the experiments. Mice were treated with temsirolimus (1 mg/kg as a single i.p. injection) or its vehicle for 20 min, and, subsequently, with THC (10 mg/kg as a single i.p. injection) or its vehicle for 4 hr. Trehalose (Merck-Calbiochem, Barcelona, Spain) was directly added to the drinking water of the animals. Mice were given trehalose (10 g/L) or plain water ad libitum for 24 hr, and, subsequently, were treated with THC (10 mg/kg as a single i.p. injection) or its vehicle for 4 hr. Under these conditions, the addition of trehalose did not affect the volume of water that was drunk by the animals. The doses of temsirolimus and trehalose used were selected from both previous studies (*Puighermanal et al., 2013*; *Rodríguez-Navarro et al., 2010*) and pilot experiments on motor behavior.

## Viral vectors

The coding sequence of dominant-negative Raptor with a c-Myc tag (Myc-RaptorΔCT) was generated by PCR-mediated deletion of 1293 base pairs at the *C*-terminus (*Hara et al., 2002*) of wild-type Myc-tagged Raptor (Addgene, Watertown, MA; plasmid #1859) (*Koketsu et al., 2008*). Myc-RaptorΔCT or human HA-tagged p62 (Addgene; plasmid #28027) was subcloned in a CAG-DIO rAAV vector, to allow the Cre-dependent expression of the transgene, by using standard molecular biology techniques. The vectors used were of an AAV1/AAV2-mixed serotype and were generated by calcium phosphate transfection of HEK-293T cells (American Type Culture Collection, Manassas, VA) and subsequent purification, as described previously (*Bellocchio et al., 2016*). Wild-type and $D_1R$-Cre mice were injected stereotactically with the rAAV vector into the dorsal striatum. Each animal received one bilateral injection at the following coordinates (to bregma): antero-posterior +0.5, lateral ±2.0, dorso-ventral −3.0 (*Bellocchio et al., 2016*). Mice were left untreated for 4 weeks to attain transgene expression before being subjected to the behavioral tests.

## Neuronal cultures

Primary striatal neurons were obtained from 0 to 1-day-old C57BL/6N mice using a papain dissociation system (Worthington, Lakewood, NJ) (*Blázquez et al., 2015*). Striata were dissected, and cells were seeded at 200,000 cells/cm$^2$ on plates that had been pre-coated with 0.1 mg/mL poly-D-lysine, in Neurobasal medium supplemented with B27 and Glutamax. Cultures were maintained for 7 days in vitro to allow neuronal differentiation. They were subsequently incubated for 24 hr with THC (0.75 µM; THC Pharm GmbH) or HU-210 (10 nM; Tocris; Bristol, UK), alone or in combination with hydroxychloroquine (0.1 mM; Merck, Barcelona, Spain) or E64d (0.1 µM; Enzo Life Sciences, Barcelona, Spain) plus pepstatin A (10 ng/ml; Enzo Life Sciences), or vehicle [DMSO, 0.1–0.2% (v/v) final concentration], before they were fixed for immunomicroscopy. Within each neuronal preparation, incubations were conducted in triplicate for every vehicle or drug condition. The total number of experimental conditions assayed within each neuronal preparation depended on the cell yield of that particular preparation.

## Immunomicroscopy

Cells were cultured on coverslips and fixed in 4% paraformaldehyde. Coronal free-floating sections (30 µm-thick) were obtained from paraformaldehyde-perfused mouse brains. Samples were first incubated with 10% goat serum in PBS supplemented with 0.25% Triton X-100 for 1 hr at room temperature to block non-specific binding, and subsequently stained overnight at 4˚C with antibodies against LC3 (1:300; Sigma-Aldrich, St. Louis, MO, #L7543), p62 (1:250; Enzo Life Sciences, #BML-PW9860-0025), LAMP1 (1:1000; Abcam, Cambridge, UK, #ab25245), DARPP32 (1:700; BD

Biosciences, Franklin Lakes, NJ, #611520), phosphorylated ribosomal protein S6-S235/S236 [1:300; Cell Signaling, Danvers, MA, #2211; similar data were obtained in pilot experiments when an antibody directed against two other mTORC1/S6K-phosphorylated residues of ribosomal protein S6 (S240/S244) was used (1:800; Cell Signaling, #5364)] or c-Myc (1:500; Sigma-Aldrich, #11-667-149-001), followed by the corresponding Alexa Fluor-conjugated secondary antibodies (1:500; 1.5 hr, room temperature, darkness; Invitrogen, Madrid, Spain, #A-11001, #A-11005, #A-21235; #A-11008, #A-11012, #A-21244). Three washes (20 min each) with 1% goat serum in PBS supplemented with 0.25%Triton X-100 were conducted both between antibody incubations and before sample mounting. Nuclei were visualized with DAPI. Analysis of marker-protein immunoreactivity in the dorsal striatum was conducted in a 1-in-10 series per animal (from bregma +1.5 to −0.5 coronal coordinates). A total of 6–8 sections (comprising 2–3 fields per section) was analyzed per mouse brain. For LC3, data were calculated as number of cells with three or more LC3-positive dots. For p62, data were calculated as immunofluorescence intensity. For DARPP32, LAMP1, c-Myc and phosphorylated ribosomal protein S6, data were calculated as number of positive cells. For tdTomato and EGFP fluorescence in *Drd1a-tdTomato/Drd2-EGFP* mice, data were calculated as number of positive cells. Confocal fluorescence images were acquired using TCS-SP8 (Leica Application suite X, LASX) software and a SP8 AOBS microscope (Leica, Wetzlar, Germany). Inclusive fluorescence thresholds were set at an average of 105 (low) and 255 (high). Images were analyzed with ImageJ software (NIH, Bethesda, MA). All immunomicroscopy analyses relied on an unbiased quantification of ImageJ-positive pixels, and were conducted in a blinded manner for genotype, pharmacological treatment and viral injection.

## Western blot

Western blot experiments were conducted with antibodies raised against LC3 (1:4000; Sigma-Aldrich, #L7543), p62 [1:1000; Enzo Life Sciences, #BML-PW9860-0025; similar data were obtained in pilot experiments when a different anti-p62 antibody was used (1:1000; Progen, Heidelberg, Germany, #GP62-C)], phosphorylated S6K-T389 (1:1000; Cell Signaling, #9206), total S6K (1:1000; Cell Signaling, #9202), phosphorylated ULK1-S757 (1:1000; Cell Signaling, #6888), phosphorylated ULK1-S555 (1:1000; Cell Signaling, #5869), total ULK1 (1:1000; Cell Signaling, #8054) or β-actin (1:4000, Sigma-Aldrich, #A5441), followed by the corresponding HRP-linked secondary antibodies (1:5000; GE-Healthcare, Madrid, Spain, #NA931; GE-Healthcare, #NA934; Invitrogen, #A18769), as appropriate. Densitometric analysis was performed with Image J software (NIH).

## Motor behavior

Motor coordination analysis (RotaRod test) was conducted with acceleration from 4 to 40 r.p.m. over a period of 600 s in an LE8200 device (Harvard Apparatus, Barcelona, Spain) (*Blázquez et al., 2011*). Any mouse remaining on the apparatus after 600 s was removed, and its time was scored as 600 s. RotaRod performance was evaluated in two phases. First, before any pharmacological treatment, naive mice were tested on three consecutive days, for three trials per day, with a rest period of 40 min between trials. Data from the three trials conducted on the first day were not used in statistical analyses, as they merely reflect the initial contact of the animal with the RotaRod device (*Hockly et al., 2003*). Data from the three trials conducted on the second day plus the three trials conducted on the third day were averaged for each animal, so constituting the herein referred to as 'pre-treatment' RotaRod performance. Second, on the day of the pharmacological experiment, 4 hr after vehicle or drug treatment, mice were tested for three trials, with a rest period of 40 min between trials. Data from these three trials were averaged for each animal, so constituting the herein referred to as 'post-treatment' RotaRod performance. Hence, for each animal, its 'post-treatment' RotaRod performance was compared with its 'pre-treatment' RotaRod performance. Motor activity analysis was conducted in an automated actimeter consisting of a 22.5 × 22.5 cm area with 16 surrounding infrared beams coupled to a computerized control unit (ActiTrack; Panlab, Barcelona, Spain) (*Blázquez et al., 2011*). Four hours after vehicle or drug treatment, animals were recorded once for a period of 10 min, in which total distance travelled (cm), overall activity (counts), resting time (s), fast movements (counts) and stereotypic movements (counts) were measured.

## Statistics

Unless otherwise specified, data are presented as mean ± SEM of the number of animals or independent neuronal preparations indicated in each case. Statistical comparisons were made by unpaired Student $t$-test, or by ANOVA followed by Tukey's multiple comparisons test, as indicated in each figure legend. For clarity, only p values lower than 0.05 were considered statistically significant. The source data files include all raw numerical data as well as further details of the statistical analyses, which were carried out with GraphPad Prism 8.0 software (San Diego, CA). Power analysis was conducted with IBM SPSS software (IBM France, Bois-Colombes, France).

## Acknowledgements

This work was supported by the Spanish *Ministerio de Ciencia, Innovación y Universidades* (MCIU/FEDER; grants SAF2015-64945-R and RTI2018-095311-B-I00 to MG, and SAF2016-78666-R to JAR-N). AR-C was supported by a contract from the Spanish *Ministerio de Ciencia, Innovación y Universidades* (MCIU/FEDER; *Formación de Personal Investigador* Program). RB-G was supported by a contract from the Spanish *Ministerio de Ciencia, Innovación y Universidades* (MCIU/FEDER; *Juan de la Cierva* Program). GM and LB were supported by INSERM. GM was also supported by *Fondation pour la Recherche Medicale* (grants DRM20101220445 and DPP20151033974), Human Frontiers Science Program (grant RGP0036/2014), European Research Council (Endofood grant ERC-2010-StG-260515, CannaPreg grant ERC-2014-PoC-640923, and MiCaBra grant ERC-2018-AdG-786467), *Region Nouvelle Aquitaine*, and *Agence Nationale de la Recherche* (ANR Blanc MitObesity grant ANR-18-CE14-0029-01, ANR LabEx BRAIN grant ANR-10-LABX-0043, ANR Blanc ORUPS grant ANR-16-CE37-0010-01, and ANR Blanc CaCoVi grant ANR 18-CE16-0001-02). LB was also supported by *Fondation pour la Recherche Medicale* (grant ARF20140129235) and *Agence Nationale de la Recherche* (ANR Blanc mitoCB1-fat grant ANR-19-CE14). We are indebted to E García-Taboada, L Rivera, A Gaudioso, P García-Rozas, and MJ Asensio for expert laboratory assistance; D Gonzales, N Aubailly, and all the personnel from the animal facilities of the NeuroCentre Magendie; M Biguerie from the technical service of the NeuroCentre Magendie; all the personnel from the Bordeaux Imaging Center; and V Morales for continuous help.

## Additional information

### Funding

| Funder | Grant reference number | Author |
|---|---|---|
| Spanish Ministerio de Ciencia, Innovación y Universidades | SAF2015-64945-R | Manuel Guzmán |
| Spanish Ministerio de Ciencia, Innovación y Universidades | RTI2018-095311-B-I00 | Manuel Guzmán |
| Spanish Ministerio de Ciencia, Innovación y Universidades | SAF2016-78666-R | José A Rodríguez-Navarro |
| Fondation pour la Recherche Médicale | DRM20101220445 | Giovanni Marsicano |
| Fondation pour la Recherche Medicale | DPP20151033974 | Giovanni Marsicano |
| Human Frontier Science Program | RGP0036/2014 | Giovanni Marsicano |
| H2020 European Research Council | ERC-2010-StG-260515 | Giovanni Marsicano |
| H2020 European Research Council | ERC-2014-PoC-640923 | Giovanni Marsicano |
| H2020 European Research Council | ERC-2018-AdG-786467 | Giovanni Marsicano |
| Region Nouvelle Aquitaine | | Giovanni Marsicano |

| Agence Nationale de la Recherche | ANR-16-CE37-0010-01 | Giovanni Marsicano |
| Agence Nationale de la Recherche | ANR 18-CE16-0001-02 | Giovanni Marsicano |
| Fondation pour la Recherche Médicale | ARF20140129235 | Luigi Bellocchio |
| Agence Nationale de la Recherche | ANR-10-LABX-0043 | Giovanni Marsicano |
| Agence Nationale de la Recherche | ANR-18-CE14-0029-01 | Giovanni Marsicano |
| Agence Nationale de la Recherche | ANR-19-CE14 | Luigi Bellocchio |

The funders had no role in study design, data collection and interpretation, or the decision to submit the work for publication.

## Author contributions
Cristina Blázquez, Data curation, Software, Formal analysis, Validation, Investigation, Visualization, Methodology, Writing - review and editing; Andrea Ruiz-Calvo, Raquel Bajo-Grañeras, Data curation, Software, Formal analysis, Validation, Investigation, Visualization, Methodology; Jérôme M Baufreton, Conceptualization, Resources, Data curation, Software, Formal analysis, Validation, Investigation, Methodology; Eva Resel, Data curation, Investigation, Methodology, Project administration; Marjorie Varilh, Antonio C Pagano Zottola, Yamuna Mariani, Astrid Cannich, Data curation, Formal analysis, Validation, Investigation, Methodology; José A Rodríguez-Navarro, Conceptualization, Resources, Data curation, Formal analysis, Validation, Investigation, Methodology, Writing - review and editing; Giovanni Marsicano, Ismael Galve-Roperh, Conceptualization, Resources, Supervision, Funding acquisition, Methodology, Project administration, Writing - review and editing; Luigi Bellocchio, Conceptualization, Resources, Data curation, Software, Formal analysis, Supervision, Funding acquisition, Validation, Investigation, Visualization, Methodology, Project administration, Writing - review and editing; Manuel Guzmán, Conceptualization, Resources, Formal analysis, Supervision, Funding acquisition, Visualization, Methodology, Writing - original draft, Project administration, Writing - review and editing

## Author ORCIDs
Jérôme M Baufreton (ID) https://orcid.org/0000-0002-2623-6375
Ismael Galve-Roperh (ID) http://orcid.org/0000-0003-3501-2434
Manuel Guzmán (ID) https://orcid.org/0000-0001-7475-118X

## Ethics
Animal experimentation: All the experimental procedures were performed in accordance with the guidelines and with the approval of the Animal Welfare Committee of Universidad Complutense de Madrid and Comunidad de Madrid (PROEX 209/18), and in accordance with the directives of the European Commission (2010/63/EU). Adequate measures were taken to minimize pain and discomfort of the animals, as well as the number of animals used in the experiments, on the basis of the 3Rs (replacement, reduction and refinement) principle.

## Decision letter and Author response
Decision letter https://doi.org/10.7554/eLife.56811.sa1
Author response https://doi.org/10.7554/eLife.56811.sa2

# Additional files

## Supplementary files
- Transparent reporting form

## Data availability

All data generated or analysed during this study are included in the manuscript and supporting files. Source data files have been provided for Figures 1 through 6.

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
