## [Decision Letter]

**Acceptance summary:**

This study demonstrates the involvement of autophagy, a regulated cell mechanism that removes unnecessary components, in the impairment in motor coordination observed following administration of THC (one of the main psychoactive components of marijuana). The findings also reveal which cells within motor pathways of the basal ganglia are responsible for this effect.

**Decision letter after peer review:**

Thank you for submitting your article "Inhibition of striatonigral autophagy as a link between cannabinoid intoxication and impairment of motor coordination" for consideration by *eLife*. Your article has been reviewed by three peer reviewers, and the evaluation has been overseen by a Reviewing Editor and Kate Wassum as the Senior Editor. The following individuals involved in review of your submission have agreed to reveal their identity: Cece Hillard (Reviewer #1); Nephi Stella (Reviewer #2).

The reviewers have discussed the reviews with one another and the Reviewing Editor has drafted this decision to help you prepare a revised submission.

Summary:

This study demonstrates the involvement of autophagy in the THC-triggered impairment in motor coordination and that the mechanism occurs specifically in striatal D_1_ MSNs. The experimental design is thorough and uses validated pharmacological and genetic tools in established rodent model systems. The technical approaches are sound, the manuscript clearly written, and conclusions and interpretations based on convincing results. The study is likely to have an important impact on the field of cannabinoid research and neuroscience in general, providing an important foundation for future studies. The report's main shortcomings are that the data (with one exception) come from a single cannabinoid ligand (given at a single dose and analyzed at a single time point after administration), that only a single (albeit well-established) behavioral measure is examined, and that only male animals were studied. As such the paper requires significant tempering of the conclusions and a thorough acknowledgement of the study's aforementioned shortcomings.

1) The complete data set needs to be provided. In particular, there are multiple instances of "representative data" being shown. Bar/dot plots of the total data sets should be provided. This includes Figures 3B, 4B, 5B, 5E, 6A and B (immuno data).

2) Figure 1: For the analysis of LC3-II by western blots, in some cases, it looks as though the LC3-I band density is changing with the LC3-II bands. At a minimum, the authors should measure and report the effects of THC on LC3-I density and should consider looking at the ratio of II/I. Also, for the p62 blots, the expression is much higher in the cortex, hippocampus and cerebellum than in the striatum in the vehicle treated samples. This raises the possibility that the lack of THC effect in these tissues is a matter of signal to noise. Please compare blots that had closer to equivalent p62 band densities in the vehicle treated groups in order to conclude whether there is a brain regional effect of THC.

3) Please use a two-way ANOVA for Figure 2 (the comparisons between THC and HU).

4) Materials and methods. How was the IHC data gathered and quantified? Were individuals scoring images or was an unbiased quantification of positive pixels used (e.g. with Image J), or another approach? A detailed description of the counting methodology and parameters is required for this section.

5) Materials and methods. The rotarod results are difficult to interpret because methodological details and data calculation specifics are missing. Considering that the mice were tested for their RotaRod performance 4 h after acute THC treatment, I understand that the 3rd RotaRod trial was measured approximately 5 h 30 min after THC treatment. Why was the first trial excluded? Does one injection of THC not impair RotaRod? Was the THC-triggered impaired in RotaRod performance more pronounced during the 3rd trial compared to the2nd trial? Was the average THC response of the RotaRod trials during first day different than the 2nd and 3rd day? Was more than one injection necessary to get a pronounced THC impaired RotaRod 4-5 h after TTT? Please include day by day data and analysis to determine the precise number of injections (on average) required to trigger the autophagy response.

6) It is striking that an impaired locomotor activity and motor coordination (RotaRod) is still detectable 4-5 h after i.p. injection in light of THC's PK profile. In the classic cannabinoid tetrad test, impaired locomotion is typically measured 30-60 min after THC injection. Please provide a detailed explanation for the prolonged impairment exerted by THC on RotaRod performance observed at 4-5 h after treatment.

7) Figure 3A: RotaRod and following RotaRod histograms: The Y axis legend and units are unclear. Should this be labeled "Motor coordination" or "Latency to fall"? Should units be in seconds or % of vehicle performance? What is basal? Is it vehicle treated? Please label accordingly. Currently, results are expressed as "relative to basal": why is the vehicle group not at 1?

8) The authors need to justify the lack of inclusion of a CB_1_ receptor selective antagonist or inverse agonist (in addition of their use of site-selective CB_1_R knock-out mice), as many things can happen in a genetically modified mouse, not all of which we the authors have control upon.

9) The authors need to convincingly discuss the limitations of using a single fairly elevated dose (10 mg/kg), which far exceeds the circulating THC plasma levels observed in cannabis-intoxicated humans.

10) In light of findings that first-pass conversion of THC to THC-OH is several times higher in female than male mice, the authors need to justify the use of male animals only because this polymorphism is likely to underlie differences in the observed effects.

11) Please ensure full reporting of your statistical analyses in your results, including F and t statistics, degrees of freedom, exact p values, etc.

---

## [Author Response]

Summary:This study demonstrates the involvement of autophagy in the THC-triggered impairment in motor coordination and that the mechanism occurs specifically in striatal D_1_ MSNs. The experimental design is thorough and uses validated pharmacological and genetic tools in established rodent model systems. The technical approaches are sound, the manuscript clearly written, and conclusions and interpretations based on convincing results. The study is likely to have an important impact on the field of cannabinoid research and neuroscience in general, providing an important foundation for future studies. The report's main shortcomings are that the data (with one exception) come from a single cannabinoid ligand (given at a single dose and analyzed at a single time point after administration), that only a single (albeit well-established) behavioral measure is examined, and that only male animals were studied. As such the paper requires significant tempering of the conclusions and a thorough acknowledgement of the study's aforementioned shortcomings.

We would like to thank the reviewers very much indeed for their positive and constructive comments, which we frankly believe have helped to improve the quality of our study. As indicated, we have explicitly acknowledged the aforementioned shortcomings of the study in the manuscript (see Discussion, paragraph one). Nonetheless, we also provide in this point-by-point response some substantiation and improvement to those issues (see below).

Revisions for this paper:1) The complete data set needs to be provided. In particular, there are multiple instances of "representative data" being shown. Bar/dot plots of the total data sets should be provided. This includes Figures 3B, 4B, 5B, 5E, 6A and B (immuno data).

We are sorry for this fault. As requested, we provide the complete data sets for those Western blot and immunofluorescence experiments, including the corresponding dot plots and statistical analyses (see Figures 3-6, as well as their legends and source data files).

2) Figure 1: For the analysis of LC3-II by western blots, in some cases, it looks as though the LC3-I band density is changing with the LC3-II bands. At a minimum, the authors should measure and report the effects of THC on LC3-I density and should consider looking at the ratio of II/I. Also, for the p62 blots, the expression is much higher in the cortex, hippocampus and cerebellum than in the striatum in the vehicle treated samples. This raises the possibility that the lack of THC effect in these tissues is a matter of signal to noise. Please compare blots that had closer to equivalent p62 band densities in the vehicle treated groups in order to conclude whether there is a brain regional effect of THC.

As requested, we show the data on the effect of THC on LC3-I levels and LC3-II/LC3-I ratios, which further support our previous data on LC3-II/β-actin ratios (Figure 1A). We have also evaluated the basal levels of LC3-I and p62 in the four brain regions under study, and no significant differences were found (Figure 1B). In addition, we provide less exposed blots. All this new information has been included in Figure 1, together with its legend and source data file, as well as in Results (paragraph one).

3) Please use a two-way ANOVA for Figure 2 (the comparisons between THC and HU).

Done (see legend and source data file for Figure 2).

4) Materials and methods. How was the IHC data gathered and quantified? Were individuals scoring images or was an unbiased quantification of positive pixels used (e.g. with Image J), or another approach? A detailed description of the counting methodology and parameters is required for this section.

We have specified that all immunomicroscopy analyses relied on an unbiased quantification of ImageJ-positive pixels, and were conducted in a blinded manner for genotype, pharmacological treatment and viral injections (Materials and methods, subsection “Immunomicroscopy”). We have also included further experimental details of the immunomicroscopy procedures (Materials and methods).

5) Materials and methods. The rotarod results are difficult to interpret because methodological details and data calculation specifics are missing. Considering that the mice were tested for their RotaRod performance 4 h after acute THC treatment, I understand that the 3rd RotaRod trial was measured approximately 5 h 30 min after THC treatment. Why was the first trial excluded? Does one injection of THC not impair RotaRod? Was the THC-triggered impaired in RotaRod performance more pronounced during the 3rd trial compared to the2nd trial? Was the average THC response of the RotaRod trials during first day different than the 2nd and 3rd day? Was more than one injection necessary to get a pronounced THC impaired RotaRod 4-5 h after TTT? Please include day by day data and analysis to determine the precise number of injections (on average) required to trigger the autophagy response.

We are sorry for the confusion that the wording of that section may have caused in the reviewers (see also point 7 below). This misinterpretation comes essentially from what “pre-treatment” and “post-treatment” RotaRod performance means for each animal. We have carefully clarified this issue now in the “Motor behaviour” subsection of the Materials and methods as follows:

“RotaRod performance was evaluated in two phases. […] Hence, for each animal, its “post-treatment” RotaRod performance was compared with its “pre-treatment” RotaRod performance.”

So, regarding the specific questions raised by the reviewers:

– Yes, the 3rd post-treatment trial was measured approximately 5 h 30 min after THC treatment.

– No, the first post-treatment trial was not excluded. As explained above, what we excluded were the trials conducted on the 1st pre-treatment day, as they merely reflect the initial contact of the animal with the RotaRod device.

– Yes, one single THC injection (as conducted throughout this study) indeed impaired (post-treatment) RotaRod performance.

– The THC-triggered impairment of RotaRod performance was similar during the 2nd and the 3rd (post-treatment) trials.

– Those three consecutive days correspond to the pre-treatment RotaRod trials, so prior to THC treatment.

– No, as mentioned above one single THC injection was sufficient to get a pronounced THC-induced (post-treatment) impairment of RotaRod performance.

– The pre-treatment and post-treatment data are detailed for each individual animal in the corresponding source data files.

6) It is striking that an impaired locomotor activity and motor coordination (RotaRod) is still detectable 4-5 h after i.p. injection in light of THC's PK profile. In the classic cannabinoid tetrad test, impaired locomotion is typically measured 30-60 min after THC injection. Please provide a detailed explanation for the prolonged impairment exerted by THC on RotaRod performance observed at 4-5 h after treatment.

We conducted the RotaRod test 4 hours after THC treatment precisely to avoid the strong distortion that acute catalepsy and motor impairment would produce on motor coordination if measured at shorter times. It is true that THC may disappear from the animal’s bloodstream by 1-2 hours after i.p. injection, but it is retained for longer time periods by – and hence it remains bioactive in – fatty tissues such as the brain. We have assessed the “cannabinoid tetrad” for many years in our labs and we consistently find that the remarkable hypolocomotor effect of 10 mg/kg THC, which – as the reviewers well say – peaks shortly after injection, declines progressively but still persists moderately 4 hours after injection (as in this study; see Figure 3—figure supplement 1 and Figure 4—figure supplement 1). Our THC administration protocol of one single i.p. injection of THC at 10 mg/kg is identical to that used by the Ozaita and Maldonado lab in Puighermanal et al., 2013, which reported a persistence of THC-evoked hypolocomotion, anxiety and analgesia 4 hours after acute injection. We have acknowledged the latter study in Results (paragraph one).

7) Figure 3A: RotaRod and following RotaRod histograms: The Y axis legend and units are unclear. Should this be labeled "Motor coordination" or "Latency to fall"? Should units be in seconds or % of vehicle performance? What is basal? Is it vehicle treated? Please label accordingly. Currently, results are expressed as "relative to basal": why is the vehicle group not at 1?

As suggested, we have re-labeled the y-axes of all RotaRod histograms with “Motor coordination”. We agree that this makes data representation clearer. Again, we are sorry for the confusion that our description of the RotaRod experiments may have caused to the reviewers. As mentioned in point 4 above, for each animal its “post-treatment” RotaRod performance was compared with its “pre-treatment” RotaRod performance, as it is widely reported in the field. “Basal” did not stand for “vehicle-treated,” but for “pre-treatment”. We understand that this may have been puzzling, and we apologize for it. So, we have replaced “basal” with “pre-treatment” throughout the text, figures and source data files. Hence, a value of “post-treatment” relative to “pre-treatment” (or “basal” in the former manuscript version) higher than 1 in the vehicle-treated, control mice simply denotes that, as it is well known in the field, the animals tend to improve their performance in the RotaRod test after consecutive trials.

8) The authors need to justify the lack of inclusion of a CB_1_ receptor selective antagonist or inverse agonist (in addition of their use of site-selective CB_1_R knock-out mice), as many things can happen in a genetically modified mouse, not all of which we the authors have control upon.

We agree with the reviewers. We have therefore evaluated whether the THC-induced impairment of motor coordination is affected by rimonabant. For this purpose, wild-type mice were injected with rimonabant (3 mg/kg, i.p.) or vehicle, and, 20 min later, with THC (10 mg/kg, i.p.) or vehicle. Four hours later, the RotaRod test was conducted. As shown in new Figure 5—figure supplement 1 (see also Results, subsection “Cannabinoid CB_1_ receptors located on the direct pathway, but not on cortical projections, are required for the THC-induced impairment of striatal autophagy and motor coordination in vivo”; Materials and methods, subsection “Drug treatments in vivo”; and figure legend), the THC-induced decrease of RotaRod performance was fully abrogated by rimonabant. Hence, this observation provides further support to the use of conditional CB_1_ receptor knockout animals to dissect the contribution of particular CB_1_ receptor subpopulations to the observed effects (Figure 5).

9) The authors need to convincingly discuss the limitations of using a single fairly elevated dose (10 mg/kg), which far exceeds the circulating THC plasma levels observed in cannabis-intoxicated humans.

This interesting question comes out very often in translational cannabinoid research. We agree that the dose of 10 mg/kg, as well alleged by the reviewers, is “fairly elevated”. However, we do not consider it a too high dose for a mouse. In fact, this dose is very frequently used to assess acute behavioral effects of THC as the “cannabinoid tetrad” (see, for example, Metna-Laurent et al., 2017). In our hands, a THC dose of 3 mg/kg produces mild effects in the “cannabinoid tetrad”, while a THC dose of 20 mg/kg leads to very strong cataleptic and motor-impairing effects that would preclude any neat evaluation of motor coordination. It should also be born in mind that, when extrapolating an animal dose to the potentially equivalent human dose, pharmacological and toxicological endpoints for systemically-administered drugs are usually assumed to scale much closer between animal species when doses are normalized to body surface area rather than to body weight. Thus, when applying these conversion factors for the definition of human equivalent doses from animal experiments (see, for example, FDA guidelines in https://www.fda.gov/regulatory-information/search-fda-guidance-documents/estimating-maximum-safe-starting-dose-initial-clinical-trials-therapeutics-adult-healthy-volunteers), an experimental dose of 10 mg/kg THC in mice would translate into 0.8 mg/kg THC in humans, which equals a total mass of 56 mg THC for a 70-kg person. This would be in turn equivalent to an estimated 0.5-0.6 g of consumed cannabis product with ~10% THC content, which falls within the range of heavy cannabis use. We frankly believe that these pieces of information support a reasonable choice of dosing for the THC administration experiments included in our study (see Results, paragraph one).

10) In light of findings that first-pass conversion of THC to THC-OH is several times higher in female than male mice, the authors need to justify the use of male animals only because this polymorphism is likely to underlie differences in the observed effects.

We agree with the reviewers. We have therefore evaluated whether the THC-induced impairment of motor coordination found in male mice also occurs in female animals. For this purpose, wild-type adult C57BL/6N female mice were injected with THC (10 mg/kg, i.p.) or vehicle. Four hours later, the RotaRod test was conducted as described in the Materials and methods section. As shown in the accompanying figure, THC decreased RotaRod performance by 40% (*n* = 5 animals per group; ***p* = 0.0035 from vehicle-treated group by unpaired Student *t*-test; *t* = 4.078; *F*_(4,4)_ = 2.018; power = 0.155). This effect is well equivalent to that exerted on male mice under identical experimental conditions (see Figures 3 through 6 in the manuscript). Hence, although we cannot rule out that certain differences may occur between male and female mice in other parameters studied, an overt absence of sex dimorphism is evident in the main behavioral trait on which our work relies.

**Author response image 1. sa2fig1:** 

**Author response table 1. resptable1:** 

Motor coordination (time to fall, s)
**Raw Data**	Vehicle	THC
	Pre-treatment	Post-treatment	Post-treatment/Pre-treatment	Pre-treatment	Post-treatment	Post-treatment/Pre-treatment
	93	101	1.08	113	71	0.63
	91	131	1.43	113	112	0.99
	79	122	1.54	101	105	1.05
	72	130	1.81	64	48	0.75
	100	143	1.44	80	80	1.00
**Mean**	87	125	1.46	94	83	0.88
**SEM**	5.05	6.96	0.12	9.66	11.62	0.08
**CI**	72.98	106.10	1.14	67.38	50.92	0.66
101.00	144.70	1.78	121.00	115.50	1.11

11) Please ensure full reporting of your statistical analyses in your results, including F and t statistics, degrees of freedom, exact p values, etc.

Done. All that information is collected in the source data files.